# In-Context Learning with Representations: Contextual Generalization of Trained Transformers

**Tong Yang**[*]  **Yu Huang**[†]  **Yingbin Liang**[‡]  **Yuejie Chi**[§]
CMU           UPenn          OSU              CMU

## Abstract

In-context learning (ICL) refers to a remarkable capability of pretrained large language models, which can learn a new task given a few examples during inference. However, theoretical understanding of ICL is largely under-explored, particularly whether transformers can be trained to generalize to unseen examples in a prompt, which will require the model to acquire contextual knowledge of the prompt for generalization. This paper investigates the training dynamics of transformers by gradient descent through the lens of non-linear regression tasks. The contextual generalization here can be attained via learning the template function for each task in-context, where all template functions lie in a linear space with $m$ basis functions. We analyze the training dynamics of one-layer multi-head transformers to in-contextly predict unlabeled inputs given partially labeled prompts, where the labels contain Gaussian noise and the number of examples in each prompt are not sufficient to determine the template. Under mild assumptions, we show that the training loss for a one-layer multi-head transformer converges linearly to a global minimum. Moreover, the transformer effectively learns to perform ridge regression over the basis functions. To our knowledge, this study is the first provable demonstration that transformers can learn contextual (i.e., template) information to generalize to both unseen examples and tasks when prompts contain only a small number of query-answer pairs.

## 1 Introduction

Transformers [Vaswani et al., 2017] have achieved tremendous successes in machine learning, particularly in natural language processing, by introducing self-attention mechanisms that enable models to capture long-range dependencies and contextualized representations. In particular, these self-attention mechanisms endow transformers with remarkable in-context learning (ICL) capabilities, allowing them to adapt to new tasks or domains by simply being prompted with a few examples that demonstrate the desired behavior, without any explicit fine-tuning or updating of the model's parameters [Brown et al., 2020].

A series of papers have empirically studied the underlying mechanisms behind in-context learning in transformer models [Garg et al., 2022, Von Oswald et al., 2023, Wei et al., 2023, Olsson et al., 2022, Xie et al., 2021, Chen and Zou, 2024, Agarwal et al., 2024], which have shown that transformers can predict unseen examples after being prompted on a few examples. The pioneering work of

---

[*]Department of Electrical and Computer Engineering, Carnegie Mellon University; email: `tongyang@andrew.cmu.edu`.

[†]Department of Statistics and Data Science, Wharton School, University of Pennsylvania; email: `yuh42@wharton.upenn.edu`.

[‡]Department of Electrical and Computer Engineering, The Ohio State University; email: `liang.889@osu.edu`.

[§]Department of Electrical and Computer Engineering, Carnegie Mellon University; email: `yuejiechi@cmu.edu`.

38th Conference on Neural Information Processing Systems (NeurIPS 2024).

Garg et al. [2022] showed empirically that transformers can be trained from scratch to perform in-context learning of simple function classes, providing a theoretically tractable in-context learning framework. Following this well-established framework, several works have investigated various properties of in-context learning in transformers. For instance, studies have explored generalization and stability [Li et al., 2023], expressive power [Bai et al., 2024, Akyürek et al., 2022, Giannou et al., 2023], causal structures [Nichani et al., 2024, Edelman et al., 2024], statistical properties [Xie et al., 2021, Jeon et al., 2024], to name a few.

In particular, analysis from an optimization perspective can provide valuable insights into how these models acquire and apply knowledge that enable in-context learning. A few works [Huang et al., 2023, Chen et al., 2024, Li et al., 2024, Nichani et al., 2024] thus studied the training dynamics of shallow transformers with softmax attention in order to in-context learn simple tasks such as linear regression [Huang et al., 2023, Chen et al., 2024], binary classification tasks [Li et al., 2024], and causal graphs [Nichani et al., 2024]. Their theoretical analyses illuminated how transformers, given an arbitrary query token, learn to *directly* apply the answer corresponding to it from the query-answer pairs that appear in each prompt. Therefore, they all require the sequence length of each prompt to be large enough so that all query-answer pairs have been seen in each prompt with sufficiently high probability, whereas practical prompts are often too short to contain many query examples. This suggests that in-context learning can exploit *inherent contextual* information of the prompt to generalize to *unseen* examples, which further raise the following intriguing theoretical question:

*How do transformers learn contextual information from more general function classes to predict unseen examples given prompts that contain only partial examples?*

Since our paper studies ICL of non-linear function regression, the function mapping (which we also term as "template") naturally serves as the "contextual information" that can be learned for generalization to unseen examples. When each prompt contains only a small number of (noisy) examples, the template that generates the labels may be *underdetermined*, i.e., multiple templates could generate the same labels in the prompt. Such an issue of underdetermination further raises a series of intriguing questions, such as:

*When the template that generates a prompt is underdetermined, what is the transformer's preference for choosing the template and how good is such a choice?*

## 1.1 Our contributions

In this paper, we answer the above questions by analyzing the training dynamics of a one-layer transformer with multi-head softmax attention through the lens of non-linear regression tasks. In our setting, the template function for each task lies in the linear space formed by $m$ nearly-arbitrary basis functions that capture representation (i.e., features) of data. Our goal is to provide insights on how transformers trained by gradient descent (GD) acquire template information from more general function classes to generalize to unseen examples and tasks when each prompt contains only a small number of query-answer pairs. We summarize our contributions are as follows.

- We first establish the convergence guarantee of a one-layer transformer with multi-head softmax attention trained with gradient descent on general non-linear regression in-context learning tasks. We assume each prompt contains only a few (i.e., partial) examples with their Gaussian noisy labels, which are not sufficient to determine the template. Under mild assumptions, we establish that the training loss of the transformer converges at a linear rate. Moreover, by analyzing the limit point of the transformer parameters, we are able to uncover what information about the basic tasks the transformer extracts and memorizes during training in order to perform in-context prediction.

- We then analyze the transformer's behavior at inference time after training, and show that the transformer chooses its generating template by performing ridge regression over the basis functions. We also provide the iteration complexity for pretraining the transformer to reach $\varepsilon$-precision with respect to its choice of the template given an arbitrary prompt at inference time. We further compare the choice of the transformer and the best possible choice over the template class and characterize how the sequence length of each prompt influences the inference time performance of the model.

- Under more realistic assumptions, our analysis framework allows us to overcome a handful of assumptions made in previous works such as large prompt length [Huang et al., 2023, Chen et al., 2024, Li et al., 2024, Nichani et al., 2024], orthogonality of data [Huang et al., 2023, Chen et al., 2024, Li et al., 2024, Nichani et al., 2024], restrictive initialization conditions [Chen et al., 2024], special structure of the transformer [Nichani et al., 2024], and mean-field models [Kim and Suzuki, 2024]. Further, the function classes we consider are a generalization of those considered in most theoretical works [Huang et al., 2023, Chen et al., 2024, Li et al., 2024, Wu et al., 2023, Zhang et al., 2023a]. We also highlight the importance of multi-head attention mechanism in this process.

To our best knowledge, this is the *first* work that analyzes how transformers learn contextual (i.e., template) information to generalize to unseen examples and tasks when prompts contain only a small number of query-answer pairs. Table 1 provides a detailed comparison with existing theoretical works in terms of settings, training analysis and generalization of in-context learning.

| Reference | nonlinear attention | multi head | task shift | GD convergence | noisy data | representation learning |
|---|---|---|---|---|---|---|
| Wu et al. [2023] | ✗ | ✗ | ✓ | ✓ | ✓ | ✗ |
| Zhang et al. [2023a] | ✗ | ✗ | ✓ | ✓ | ✓ | ✗ |
| Huang et al. [2023] | ✓ | ✗ | ✓ | ✓ | ✗ | ✗ |
| Li et al. [2024] | ✓ | ✗ | ✓ | ✓ | ✓ | ✗ |
| Chen et al. [2024] | ✓ | ✓ | ✗ | ✗ | ✓ | ✗ |
| Kim and Suzuki [2024] | ✗ | ✗ | ✓ | ✗ | ✗ | ✓ |
| Ours | ✓ | ✓ | ✓ | ✓ | ✓ | ✓ |

Table 1: Comparisons with existing theoretical works that study the learning dynamics of transformers in ICL. Here, the last column refers to the fact that the response in the regression task is generated by a linearly weighted unknown representation (feature) model.

## 1.2 Related work

**In-context learning.** Recent research has investigated the theoretical underpinnings of transformers' ICL capabilities from diverse angles. For example, several works focus on explaining the in-context learning of transformers from a Bayesian perspective [Xie et al., 2021, Ahuja et al., 2023, Han et al., 2023, Jiang, 2023, Wang et al., 2023, Wies et al., 2024, Zhang et al., 2023b, Jeon et al., 2024, Hahn and Goyal, 2023]. Li et al. [2023] analyzed the generalization and stability of transformers' in-context learning. Focusing on the representation theory, Akyürek et al. [2022], Bai et al. [2024] studied the expressive power of transformers on the linear regression task. Akyürek et al. [2022] showed by construction that transformers can represent GD of ridge regression or the closed-form ridge regression solution. Bai et al. [2024] extended Akyürek et al. [2022] and showed that transformers can implement a broad class of standard machine learning algorithms in-context. Dai et al. [2022], Von Oswald et al. [2023] showed transformers could in-context learn to perform GD.

More pertinent to our work, Guo et al. [2023] considered an ICL setting very similar to ours, where the label depends on the input through a basis of possibly complex but fixed template functions, composed with a linear function that differs in each prompt. By construction, the optimal ICL algorithm first transforms the inputs by the representation function, and then performs linear ICL on top of the transformed dataset. Guo et al. [2023] showed the existence of transformers that approximately implement such algorithms, whereas our work is from a different perspective, showing that (pre)training the transformer loss by GD will naturally yield a solution with the aforementioned desirable property characterized in Guo et al. [2023].

**Training dynamics of transformers performing ICL.** A line of work initiated by Garg et al. [2022] aims to understand the ICL ability of transformers from an optimization perspective. [Zhang et al., 2023a, Kim and Suzuki, 2024] analyzed the training dynamics of transformers with *linear* attention. Huang et al. [2023], Chen et al. [2024], Li et al. [2024] studied the optimization dynamics of one-layer softmax attention transformers performing simple in-context learning tasks, such as linear regression [Huang et al., 2023, Chen et al., 2024] and binary classification [Li et al., 2024].

Among them, Huang et al. [2023] was the first to study the training dynamics of softmax attention, where they gave the convergence results of a one-layer transformer with single-head attention on linear regression tasks, assuming context features come from an orthogonal dictionary and each token in the prompts is drawn from a multinomial distribution. In order to leverage the concentration property inherent to multinomial distributions, they require the sequence length to be much larger than the size of dictionary. Their analysis indicates that the prompt tokens that are the same as the query will have dominating attention weights, which allows the transformer to *copy-paste* the correct answer from those prompt tokens.

Li et al. [2024] studied the training of a one-layer single-head transformer in ICL on binary classification tasks. Same as Huang et al. [2023], they required the data to be pairwise orthogonal, and shared the same copy-paste mechanism as in Huang et al. [2023]. To be precise, a fraction of their context inputs needs to contain the same pattern as the query to guarantee that the total attention weights on contexts matching the query pattern outweigh those on other contexts.

Chen et al. [2024] studied the dynamics of *gradient flow* for training a one-layer multi-head softmax attention model for ICL of multi-task linear regression, where the coefficient matrix has certain spectral properties. They required the sequence length to be sufficiently large [Chen et al., 2024, Assumption 2.1], together with restrictive initialization conditions [Chen et al., 2024, Definition 3.1]. While using the copy-paste analysis framework as in Huang et al. [2023], Li et al. [2024], the attention probability vector in their work is delocalized, so that the attention is spread out to capture the information from similar tokens in regression tasks. Kim and Suzuki [2024] studied the dynamics of Wasserstein gradient flow for training a one-layer transformer with an infinite-dimensional fully-connected layer followed by a linear attention layer for ICL of linear regression, assuming infinite prompt length. Nichani et al. [2024] analyzed the optimization dynamics of a simplified two-layer transformer with gradient descent on in-context learning a latent causal graph.

**Notation.** Boldface small and capital letters denote vectors and matrices, respectively. Sets are denoted with curly capital letters, e.g., $\mathcal{W}$. We let $(\mathbb{R}^d, \|\cdot\|)$ denote the $d$-dimensional real coordinate space equipped with norm $\|\cdot\|$. $\boldsymbol{I}_d$ is the identity matrix of dimension $d$. The $\ell^p$-norm of $\boldsymbol{v}$ is denoted by $\|\boldsymbol{v}\|_p$, where $1 \leq p \leq \infty$, and the spectral norm and the Frobenius norm of a matrix $\boldsymbol{M}$ are denoted by $\|\boldsymbol{M}\|_2$ and $\|\boldsymbol{M}\|_F$, respectively. $\boldsymbol{M}^\dagger$ stands for the Moore-Penrose pseudoinverse of matrix $\boldsymbol{M}$, and $\boldsymbol{M}_{:,i}$ stands for its $i$-th column vector. We let $[N]$ denote $\{1, \ldots, N\}$, and denote $\mathbf{1}_N$ to represent the all-one vector of length $N$, and by $\mathbf{0}$ a vector or a matrix consisting of all 0's. We allow the application of functions such as $\exp(\cdot)$ to vectors or matrices, with the understanding that they are applied in an element-wise manner. We use $\boldsymbol{e}_i$ to denote the one-hot vector whose $i$-th entry is 1 and the other entries are all 0.

## 2 Problem Setup

**In-context learning with representation.** We consider ICL of regression with unknown representation, similar to the setup introduced in Guo et al. [2023]. To begin, let $f : \mathbb{R}^d \to \mathbb{R}^m$ be a fixed representation map that $f(\boldsymbol{x}) = (f_1(\boldsymbol{x}), \cdots, f_m(\boldsymbol{x}))^\top$ for any $\boldsymbol{x} \in \mathbb{R}^d$. The map $f$ can be quite general, which can be regarded as a feature extractor that will be learned by the transformer. We assume that each ICL task corresponds to a map $\boldsymbol{\lambda}^\top f(\cdot)$ that lies in the linear span of those $m$ basis functions in $f(\cdot)$, where $\boldsymbol{\lambda}$ is generated according to the distribution $\mathcal{D}_{\boldsymbol{\lambda}}$. Thus, for each ICL instance, the (noisy) label of an input $\boldsymbol{v}_k$ ($\forall k \in [K]$) is given as

$$y_k = \boldsymbol{\lambda}^\top (f(\boldsymbol{v}_k) + \boldsymbol{\epsilon}_k), \qquad \boldsymbol{\lambda} \sim \mathcal{D}_{\boldsymbol{\lambda}}, \quad \boldsymbol{\epsilon}_k \overset{i.i.d.}{\sim} \mathcal{N}(0, \tau \boldsymbol{I}_m) \tag{1}$$

where $\tau > 0$ is the noise level.

The goal of ICL is to form predictions on query $\boldsymbol{x}_{\text{query}}$ given in-context labels of the form (1) on a few inputs, known as *prompts*. In this paper, we use $\mathcal{V}$ to denote the *dictionary* set that contains all $K$ unit-norm *distinct* tokens, i.e., $\mathcal{V} := \{\boldsymbol{v}_1, \cdots, \boldsymbol{v}_K\} \subset \mathbb{R}^d$ with each token $\|\boldsymbol{v}_k\|_2 = 1$. We assume that each prompt $P = P_{\boldsymbol{\lambda}}$ provides the first $N$ tokens (with $N \ll K$) and their labels, and is embedded in the following matrix

$$\boldsymbol{E}^P := \begin{pmatrix} \boldsymbol{v}_1 & \boldsymbol{v}_2 & \cdots & \boldsymbol{v}_N \\ y_1 & y_2 & \cdots & y_N \end{pmatrix} := \begin{pmatrix} \boldsymbol{V} \\ \boldsymbol{y}^\top \end{pmatrix} \in \mathbb{R}^{(d+1) \times N}, \tag{2}$$

where

$$\boldsymbol{V} := (\boldsymbol{v}_1, \cdots, \boldsymbol{v}_N) \in \mathbb{R}^{d \times N} \tag{3}$$

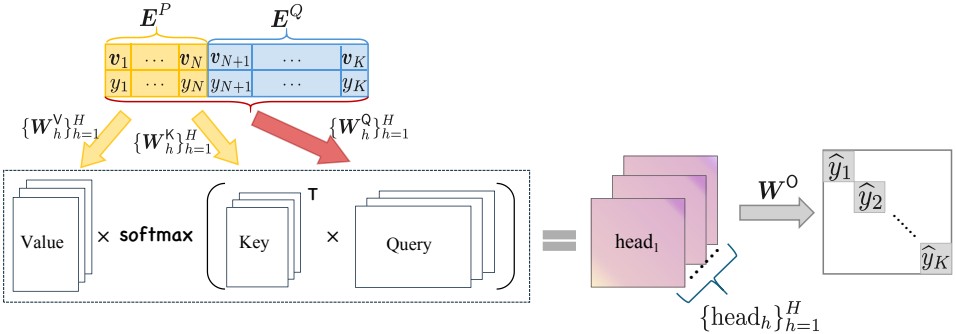

Figure 1: The structure of a one-layer transformer with multi-head softmax attention.

is the collection of prompt tokens, and $\boldsymbol{y} := (y_1, \cdots, y_N)^\top$ is the prompt label. Given the prompt as the input, the transformer predicts the labels for all the $K$ tokens $y_1, \cdots, y_K$ in the dictionary set.

**Transformer architecture.** We adopt a one-layer transformer with multi-head softmax attention [Chen et al., 2024] — illustrated in Figure 1 — to predict the labels of all the tokens in the dictionary $\mathcal{V}$, where $H$ is the number of heads. Denote the query embedding as

$$\boldsymbol{E}^Q := \begin{pmatrix} \boldsymbol{v}_{N+1} & \boldsymbol{v}_{N+2} & \cdots & \boldsymbol{v}_K \\ 0 & 0 & \cdots & 0 \end{pmatrix} \in \mathbb{R}^{(d+1)\times(K-N)}, \tag{4}$$

and denote the embedding of both the prompt and the query as $\boldsymbol{E} := (\boldsymbol{E}^P, \boldsymbol{E}^Q) \in \mathbb{R}^{(d+1)\times K}$. We define the output of each transformer head as

$$\mathrm{head}_h(\boldsymbol{E}) := \boldsymbol{W}_h^{\mathsf{V}} \boldsymbol{E}^P \cdot \mathsf{softmax}\left((\boldsymbol{E}^P)^\top (\boldsymbol{W}_h^{\mathsf{K}})^\top \boldsymbol{W}_h^{\mathsf{Q}} \boldsymbol{E}\right), \quad h \in [H], \tag{5}$$

where $\boldsymbol{W}_h^{\mathsf{Q}} \in \mathbb{R}^{d_e \times (d+1)}$, $\boldsymbol{W}_h^{\mathsf{K}} \in \mathbb{R}^{d_e \times (d+1)}$, and $\boldsymbol{W}_h^{\mathsf{V}} \in \mathbb{R}^{K \times (d+1)}$ are the query, key, and value matrices, respectively, and the softmax is applied column-wisely, i.e., given a vector input $\boldsymbol{x}$, the $i$-th entry of $\mathsf{softmax}(\boldsymbol{x})$ is given by $e^{x_i} / \sum_j e^{x_j}$. The attention map of the transformer $\mathcal{T}(\boldsymbol{E})$ is defined as

$$\mathcal{T}(\boldsymbol{E}) := \boldsymbol{W}^{\mathsf{O}} \begin{pmatrix} \mathrm{head}_1(\boldsymbol{E}) \\ \vdots \\ \mathrm{head}_H(\boldsymbol{E}) \end{pmatrix} \in \mathbb{R}^{K \times K}, \tag{6}$$

where $\boldsymbol{W}^{\mathsf{O}}$ is the output matrix. Following recent theoretical literature to streamline analysis [Huang et al., 2023, Nichani et al., 2024, Deora et al., 2023, Chen et al., 2024], we assume that the embedding matrices take the following forms:

$$\boldsymbol{W}^{\mathsf{O}} := (\boldsymbol{I}_K, \cdots, \boldsymbol{I}_K) \in \mathbb{R}^{K \times HK}, \quad \boldsymbol{W}_h^{\mathsf{V}} := (\boldsymbol{0}, \boldsymbol{w}_h) \in \mathbb{R}^{K \times (d+1)}, \tag{7a}$$

$$(\boldsymbol{W}_h^{\mathsf{K}})^\top \boldsymbol{W}_h^{\mathsf{Q}} = \begin{pmatrix} \boldsymbol{Q}_h & \boldsymbol{0} \\ \boldsymbol{0} & 0 \end{pmatrix} \in \mathbb{R}^{(d+1)\times(d+1)}, \quad \forall h \in [H], \tag{7b}$$

where $\boldsymbol{w}_h = (w_{h,1}, \cdots, w_{h,K})^\top \in \mathbb{R}^K$ and $\boldsymbol{Q}_h \in R^{d \times d}$ are trainable parameters for all $h \in [H]$.

The prediction of the labels is provided by the diagonal entries of $\mathcal{T}(\boldsymbol{E})$, which we denote by $\widehat{\boldsymbol{y}} = (\widehat{y}_1, \cdots, \widehat{y}_K) \in \mathbb{R}^K$. Note that $\widehat{y}_k$ takes the following form under our parameter specification:

$$\forall k \in [K]: \qquad \widehat{y}_k = \left\langle \boldsymbol{y}, \sum_{h=1}^{H} w_{h,k}\, \mathsf{softmax}(\boldsymbol{V}^\top \boldsymbol{Q}_h \boldsymbol{v}_k) \right\rangle. \tag{8}$$

**Training via GD.** Let $\boldsymbol{\theta} = \{\boldsymbol{Q}_h, \boldsymbol{w}_h\}_{h=1}^{H}$ denote all trainable parameters of $\mathcal{T}$. Let $\boldsymbol{\epsilon} := (\boldsymbol{\epsilon}_1, \cdots, \boldsymbol{\epsilon}_K) \in \mathbb{R}^{m \times K}$ denote the noise matrix. Given training data over ICL instances, the goal of training is to predict labels $y_k$ for all $\boldsymbol{v}_k \in \mathcal{V}$. Specifically, we train the transformer using gradient descent (GD) by optimizing the following mean-squared population loss:

$$\mathcal{L}(\boldsymbol{\theta}) := \frac{1}{2} \mathbb{E}_{\boldsymbol{\lambda}, \boldsymbol{\epsilon}} \left[ \frac{1}{K} \sum_{k=1}^{K} (\widehat{y}_k - y_k)^2 \right]. \tag{9}$$

We apply different learning rates $\eta_Q, \eta_w > 0$ for updating $\{Q_h\}_{h=1}^{H}$ and $\{w_h\}_{h=1}^{H}$, respectively, i.e., at the $t$-th ($t \geq 1$) step, we have

$$\forall h \in [H]: \quad Q_h^{(t)} = Q_h^{(t-1)} - \eta_Q \nabla_{Q_h} \mathcal{L}(\theta^{(t-1)}), \quad w_h^{(t)} = w_h^{(t-1)} - \eta_w \nabla_{w_h} \mathcal{L}(\theta^{(t-1)}), \quad (10)$$

where $\theta^{(t)} = \{Q_h^{(t)}, w_h^{(t)}\}_{h=1}^{H}$ is the parameter at the $t$-th step.

**Inference time.** At inference time, given a prompt $P = P_{\lambda}$ with $N$ examples, where $\lambda$ *may not be in the support of the generation distribution $\mathcal{D}_{\lambda}$*, the transformer applies the pretrained parameters and predicts the labels of all $K$ tokens without further parameter updating.

## 3 Theoretical Analysis

### 3.1 Training time convergence

In this section, we show that the training loss $\mathcal{L}$ converges to its minimum value at a linear rate during training, i.e., the function gap

$$\Delta^{(t)} := \mathcal{L}(\theta^{(t)}) - \inf_{\theta} \mathcal{L} \to 0, \quad t \to \infty \quad (11)$$

at a linear rate, under some appropriate assumptions.

**Key assumptions.** We first state our technical assumptions. The first assumption is on the distribution $\mathcal{D}_{\lambda}$ for generating the coefficient vector $\lambda$ of the representation maps.

**Assumption 1** (Assumption on distribution $\mathcal{D}_{\lambda}$). *We assume that in (1) each entry $\lambda_i$ is drawn independently and satisfies $\mathbb{E}[\lambda_i] = 0$ and $\mathbb{E}[\lambda_i^2] = 1$ for all $i \in [m]$.*

To proceed, we introduce the following notation:

$$Z := (f(v_1) \cdots f(v_N)) \in \mathbb{R}^{m \times N}, \quad \bar{Z} := (Z^{\top} Z + m\tau I_N)^{1/2} \in \mathbb{R}^{N \times N}, \quad \bar{f}_{\max} := \max_{i \in [N]} \|\bar{z}_i\|_2, \quad (12)$$

where $\bar{z}_i$ is the $i$-th column vector of $\bar{Z}$ for $i \in [N]$. We further define $C_k^{(t)}$ ($k \in [K]$, $t \in \mathbb{N}_+$) and $B_k^{(t)}$ as follows:

$$C_k^{(t)} := \mathsf{softmax}(V^{\top} Q_1^{(t)} v_k, \cdots, V^{\top} Q_H^{(t)} v_k) \in \mathbb{R}^{N \times H}, \qquad B_k^{(t)} = \bar{Z} C_k^{(t)} \in \mathbb{R}^{N \times H}. \quad (13)$$

To guarantee the convergence, we require the initialization of the parameters $\theta^{(0)}$ satisfies the following condition.

**Assumption 2** (Assumption on initialization). *For all $k \in [K]$, $B_k^{(0)}$ has full row rank.*

Before stating our main theorem, let us examine when the initialization condition in Assumption 2 is met. Fortunately, we only require the following mild assumption on $V$ to ensure our parameter initialization has good properties.

**Assumption 3** (Assumption on $V$). *There exists one row vector $x = (x_1, \cdots, x_N)^{\top}$ of the prompt token matrix $V$ (cf. (3)) such that $x_i \neq x_j, \forall i \neq j$.*

Assumption 3 implies that $\mathcal{V}$ has distinct tokens, i.e., $v_j \neq v_k$ when $j \neq k$. It is worth noting that Assumption 3 is the only assumption we have on the dictionary $\mathcal{V}$. In comparison, all other theoretical works in Table 1 impose somewhat unrealistic assumptions on $\mathcal{V}$. For example, Huang et al. [2023], Li et al. [2023], Nichani et al. [2024] assume that the tokens are pairwise orthogonal, which is restrictive since it implies that the dictionary size $K$ should be no larger than the token dimension $d$, whereas in practice it is often the case that $K \gg d$ [Reid et al., 2024, Touvron et al., 2023]. In addition, Chen et al. [2024], Zhang et al. [2023a], Wu et al. [2023] assume that each token is independently sampled from some Gaussian distribution, which also does not align with practical scenarios where tokens are from a fixed dictionary and there often exist (strong) correlations between different tokens.

The following proposition states that when the number of heads exceeds the number of prompts, i.e. $H \geq N$, we can guarantee that Assumption 2 holds with probability 1 by simply initializing $\{Q_h\}_{h=1}^{H}$ using Gaussian distribution.

**Proposition 1** (Initialization of $\{\boldsymbol{Q}_h\}_{h=1}^H$). *Suppose Assumptions 1, 3 hold and $H \geq N$. For any fixed $\beta > 0$, let $\boldsymbol{Q}_h^{(0)}(i,j) \overset{i.i.d.}{\sim} \mathcal{N}(0, \beta^2)$, then Assumption 2 holds almost surely.*

*Proof.* See Appendix E.1. □

**Choice of learning rates.** Define

$$\zeta_0 := \min_{k \in [K]} \left\{ \lambda_{\min} \left( \boldsymbol{B}_k^{(0)} \boldsymbol{B}_k^{(0)\top} \right) \right\}, \tag{14}$$

where $\Delta^{(0)}$ is the initial function gap (c.f. (11)). Assumption 2 indicates that $\zeta_0 > 0$. Let $\gamma$ be any positive constant that satisfies

$$\gamma \geq \zeta_0^{-5/4} \left( \frac{128\sqrt{2}}{\sqrt{2}-1} \left\| \bar{\boldsymbol{Z}} \right\|_2^2 \sqrt{H} \bar{f}_{\max} K^{3/2} \Delta^{(0)} \right)^{1/2}. \tag{15}$$

We set the learning rates as

$$\eta_Q \leq 1/L \quad \text{and} \quad \eta_w = \gamma^2 \eta_Q, \tag{16}$$

where[5]

$$L^2 = \left( 8\sqrt{2} H \sqrt{K} \frac{\|\bar{\boldsymbol{z}}\|_2^2}{\zeta_0} \sqrt{\Delta^{(0)}} + 1 + \frac{\|\boldsymbol{z}^\top \hat{\boldsymbol{z}}\|_2}{m\tau} \right)^2 \left\| \bar{\boldsymbol{Z}} \right\|_2^4 \cdot \left( \frac{8}{K} \gamma^2 + \frac{4096}{\gamma \zeta_0^2} K^2 N \Delta^{(0)} \right)$$

$$+ 2H^2 \left\| \bar{\boldsymbol{Z}} \right\|_2^4 \left( \frac{\gamma^4}{K^2} + \frac{16384}{\gamma \zeta_0^4} K^3 \left\| \bar{\boldsymbol{Z}} \right\|_2^2 \left( \Delta^{(0)} \right)^2 \right). \tag{17}$$

**Theoretical guarantee.** Now we are ready to state our first main result, regarding the training dynamic of the transformer.

**Theorem 1** (Training time convergence). *Suppose Assumptions 1, 2 hold. We let $\boldsymbol{w}_k^{(0)} = \boldsymbol{0}$ and set the learning rates as in (16). Then we have*

$$\mathcal{L}(\boldsymbol{\theta}^{(t)}) - \inf_{\boldsymbol{\theta}} \mathcal{L}(\boldsymbol{\theta}) \leq \left( 1 - \frac{\eta_w \zeta_0}{2K} \right)^t \left( \mathcal{L}(\boldsymbol{\theta}^{(0)}) - \inf_{\boldsymbol{\theta}} \mathcal{L}(\boldsymbol{\theta}) \right), \quad \forall t \in \mathbb{N}. \tag{18}$$

*Proof.* See Appendix C. □

Theorem 1, together with Proposition 1, shows that the training loss converges to its minimum value at a linear rate, under mild assumptions of the task coefficients and token dictionary. This gives the *first* convergence result for transformers with multi-head softmax attention trained using GD to perform ICL tasks (see Table 1). Our convergence guarantee (18) also indicates that the convergence speed decreases as the size $K$ of the dictionary or the number $H$ of attention heads increases, which is intuitive because training with a larger vocabulary size or number of parameters is more challenging. However, a small $H$ will limit the expressive power of the model (see Section 3.3 for detailed discussion), and we require $H \geq N$ to guarantee Assumption 2 holds, as stated in Proposition 1.

### 3.2 Inference time performance

We now move to examine the inference time performance, where the coefficient vector $\boldsymbol{\lambda}$ corresponding to the inference task may not drawn from $\mathcal{D}_{\boldsymbol{\lambda}}$. In fact, we only assume that the coefficient vector $\boldsymbol{\lambda}$ at inference time is bounded as in the following assumption.

**Assumption 4** (Boundedness of $\boldsymbol{\lambda}$ at inference time). *We assume that at inference time $\|\boldsymbol{\lambda}\|_2 \leq B$ for some $B > 0$.*

For notational simplicity, let $\boldsymbol{Z}^Q \in \mathbb{R}^{m \times (K-N)}$ denote

$$\boldsymbol{Z}^Q := (f(\boldsymbol{v}_{N+1}), \cdots, f(\boldsymbol{v}_K)) \in \mathbb{R}^{m \times (K-N)}. \tag{19}$$

The following theorem characterizes the performance guarantee of the transformer's output $\hat{\boldsymbol{y}}$ (after sufficient training) at the inference time.

---

[5] We leave a tighter, but more complicated, expression of $L$ in the appendix (cf. (61)) in the appendix and present a simplified form in the main paper for readability.

**Theorem 2** (Inference time performance). *Let $\widehat{\boldsymbol{\lambda}}$ be the solution to the following ridge regression problem:*

$$\widehat{\boldsymbol{\lambda}} := \arg\min_{\boldsymbol{\lambda}} \left\{ \frac{1}{2N} \sum_{i=1}^{N} (y_i - \boldsymbol{\lambda}^\top f(\boldsymbol{v}_i))^2 + \frac{m\tau}{2N} \|\boldsymbol{\lambda}\|_2^2 \right\}. \tag{20}$$

*Under the assumptions in Theorem 1, for any $\varepsilon > 0$ and $\delta \in (0,1)$, if the number of training iterates $T$ satisfies*

$$T \geq \frac{\log\left( B^2 \Delta^{(0)} \left( \|\boldsymbol{Z}\|_2 + \sqrt{\tau}\left( 2\sqrt{N\log(1/\delta)} + 2\log(1/\delta) + N \right)^{1/2} \right)^2 \Big/ (m\tau\varepsilon) \right)}{\log\left( 1 \big/ \left( 1 - \frac{\eta_w \zeta_0}{2K} \right) \right)}, \tag{21}$$

*then given any prompt $P$ that satisfies Assumption 4 at the inference time, with probability at least $1 - \delta$, the output of the transformer $\widehat{\boldsymbol{y}}$ satisfies*

$$\frac{1}{2K} \|\widehat{\boldsymbol{y}} - \widehat{\boldsymbol{y}}^\star\|_2^2 \leq \varepsilon, \qquad \text{with} \quad \widehat{\boldsymbol{y}}^\star := \begin{pmatrix} \boldsymbol{y} \\ \left(\boldsymbol{Z}^Q\right)^\top \widehat{\boldsymbol{\lambda}} \end{pmatrix}. \tag{22}$$

*Proof.* See Appendix D. □

In Theorem 2, (22) shows that after training, the transformer learns to output the given labels of the first $N$ tokens in each prompt, and more importantly, predicts the labels of the rest $K - N$ tokens by implementing the ridge regression given in (20). Note that Akyürek et al. [2022] studied the expressive power of transformers on the linear regression task and showed by construction that transformers can represent the closed-form ridge regression solution. Interestingly, here we show from an optimization perspective that transformers can in fact be trained to do so.

**Generalization capabilities of the pretrained transformer.** Theorem 2 captures two generalization capabilities that the pretrained transformer can have.

i) *Contextual generalization to unseen examples:* Theorem 2 suggests that the transformer exploits the *inherent contextual* information (to be further discussed in Section 3.3) of the function template in the given prompt, and can further use such information to predict the unseen tokens.

ii) *Generalization to unseen tasks:* Theorem 2 also suggests that the pretrained transformer can generalize to a function map corresponding to any $\boldsymbol{\lambda} \in \mathbb{R}^m$ at the inference time (albeit satisfying Assumption 4), which is not necessarily sampled from the support of its training distribution $\mathcal{D}_{\boldsymbol{\lambda}}$.

We note that the contextual generalization that the transformer has here is different in nature from the prediction ability shown in previous works on ICL [Huang et al., 2023, Chen et al., 2024, Li et al., 2024, Nichani et al., 2024]. Those work focuses on a setting where each prompt contains a good portion of tokens similar to the query token, allowing the transformer to *directly* use the label of the corresponding answers from the prompt as the prediction. However, in practical scenarios, prompts often contain only partial information, and our analysis sheds lights on explaining how transformers generalize to unseen examples by leveraging ridge regression to infer the underlying template.

**How does the representation dimension affect the performance?** Beyond the above discovery, several questions are yet to be explored. For instance, while we demonstrate that transformers can be trained to implement ridge regression, how good is the performance of the ridge regression itself? What is the best choice of ridge regression we could expect? How close is the transformer's choice to the best possible choice? We address these questions as follows.

Given any prompt $P$ at inference time, since there is no label information about the rest $K - N$ tokens, the best prediction we could hope for from the transformer shall be

$$\widehat{\boldsymbol{y}}^{\text{best}} := \begin{pmatrix} \boldsymbol{y} \\ \left(\boldsymbol{Z}^Q\right)^\top \widehat{\boldsymbol{\lambda}}_\tau \end{pmatrix}, \tag{23}$$

where $\boldsymbol{Z}^Q$ is defined in (19), and $\widehat{\boldsymbol{\lambda}}_\tau$ satisfies:

$$\widehat{\boldsymbol{\lambda}}_\tau := \arg\min_{\boldsymbol{\lambda}} \mathbb{E}_{\tilde{\boldsymbol{\epsilon}}} \left[ \frac{1}{2N} \sum_{i=1}^{N} (y_i - \boldsymbol{\lambda}^\top (f(\boldsymbol{v}_i) + \boldsymbol{\epsilon}_i))^2 \right]. \tag{24}$$

In other words, we hope the transformer outputs the given $N$ labels as they are. For the rest $K - N$ labels, the best we could hope for is that the transformer estimates the coefficient vector $\boldsymbol{\lambda}$ by solving the above regression problem to obtain $\widehat{\boldsymbol{\lambda}}_\tau$, and predict the $k$-th label by $\widehat{\boldsymbol{\lambda}}_\tau^\top f(\boldsymbol{v}_k)$ for $k = N + 1, \cdots, K$. Note that (24) is equivalent to the following ridge regression problem (see Lemma 4 in the appendix for its derivation):

$$\widehat{\boldsymbol{\lambda}}_\tau = \arg\min_{\boldsymbol{\lambda}} \left\{ \frac{1}{2N} \sum_{i=1}^{N} (y_i - \boldsymbol{\lambda}^\top f(\boldsymbol{v}_i))^2 + \frac{\tau}{2} \|\boldsymbol{\lambda}\|_2^2 \right\}. \tag{25}$$

The only difference between the two ridge regression problems (20) and (25) is the coefficient of the regularization term. This indicates that at the training time, the transformer learns to implement ridge regression to predict the labels of the rest $K - N$ tokens, assuming the noise level is given by $\frac{m}{N}\tau$. This observation also reflects how the sequence length $N$ affects the transformer's preference for choosing templates and its performance at inference time:

- The closer $m$ is to $N$, the closer the transformer's choice of templates is to the best possible choice, and the better the transformer's prediction will be;

- When $N < m$, the transformer tends to underfit by choosing a $\boldsymbol{\lambda}$ with small $\ell_2$-norm;

- When $N > m$, the transformer tends to overfit since it underestimates the noise level and in turn captures noise in the prediction.

## 3.3 Further interpretation

We provide more interpretation on our results, which may lead to useful insights into the ICL ability of the transformer.

**How does the transformer gain ICL ability with representations?** Intuitively speaking, our pretrained transformer gains in-context ability by extracting and memorizing some "inherent information" of all basic function maps $f_i$ ($i \in [m]$) during the training. Such information allows it to infer the coefficient vector $\boldsymbol{\lambda}$ from the provided labels in each prompt and calculate the inner product $\langle \boldsymbol{\lambda}, f(\boldsymbol{v}_k) \rangle$ to compute $y_k$ given any token $\boldsymbol{v}_k \in \mathcal{V}$ at inference time. To be more specific, the "inherent information" of all basic tasks could be described by the $N$-by-$K$ matrix $\boldsymbol{A}$ defined as follows (see also (34)):

$$\boldsymbol{A} := \left( \boldsymbol{Z}^\top \boldsymbol{Z} + m\tau \boldsymbol{I}_N \right)^{-1} \left( \boldsymbol{Z}^\top \widehat{\boldsymbol{Z}} + (m\tau \boldsymbol{I}_N, \boldsymbol{0}) \right) \in \mathbb{R}^{N \times K},$$

where $\widehat{\boldsymbol{Z}} := (f(\boldsymbol{v}_1), \cdots, f(\boldsymbol{v}_K)) = (\boldsymbol{Z}, \boldsymbol{Z}^Q) \in \mathbb{R}^{m \times K}$. During training, the transformer learns to approximate $\boldsymbol{A}_{:,k}$ by $\sum_{h=1}^{H} w_{h,k} \mathsf{softmax}(\boldsymbol{V}^\top \boldsymbol{Q}_h \boldsymbol{v}_k)$ for each $k \in [K]$.

To further elaborate, we take a closer look at the special case when the labels do not contain any noise, i.e., $\tau = 0$, and $N \geq m$. In this case, $\boldsymbol{A}$ becomes $\boldsymbol{Z}^\dagger \widehat{\boldsymbol{Z}}$, and given any prompt $P = P_{\boldsymbol{\lambda}}$, the coefficient vector $\boldsymbol{\lambda}$ could be uniquely determined from the provided token-label pairs in the prompt. It is straightforward to verify that the label of each token $\boldsymbol{v}_k$ could be represented by the inner product of the given label vector $\boldsymbol{y}$ and the $k$-th column of $\boldsymbol{Z}^\dagger \widehat{\boldsymbol{Z}}$, i.e.,

$$y_k = \left\langle \boldsymbol{y}, \boldsymbol{Z}^\dagger \widehat{\boldsymbol{Z}}_{:,k} \right\rangle. \tag{26}$$

Comparing the above equation with (8), it can be seen that in order to gain the in-context ability, the transformer needs to learn an approximation of $\boldsymbol{Z}^\dagger \widehat{\boldsymbol{Z}}_{:,k}$ by $\sum_{h=1}^{H} w_{h,k} \mathsf{softmax}(\boldsymbol{V}^\top \boldsymbol{Q}_h \boldsymbol{v}_k)$ for each $k \in [K]$.

More generally, in the proof of Theorem 2, we show that

$$\widehat{y}_k^\star = \langle \boldsymbol{y}, \boldsymbol{A}_{:,k} \rangle, \tag{27}$$

comparing which with (8) suggests that a small training error implies that $\sum_{h=1}^{H} w_{h,k} \mathsf{softmax}(\boldsymbol{V}^\top \boldsymbol{Q}_h \boldsymbol{v}_k)$ is close to $\boldsymbol{A}_{:,k}$. In fact, this is the necessary and sufficient condition for the training loss to be small. A rigorous argument is provided in Lemma 5.

**The necessity and trade-offs of multi-head attention mechanism.** Multi-head attention mechanism is essential in our setting. In fact, it is generally impossible to train a shallow transformer with only one attention head to succeed in the ICL task considered in our paper. This is because, as we have discussed above, the key for the transformer is to approximate $\boldsymbol{A}_{:,k}$ by $\sum_{h=1}^{H} w_{h,k}\mathsf{softmax}(\boldsymbol{V}^{\top}\boldsymbol{Q}_h\boldsymbol{v}_k)$ for each $k \in [K]$. If $H = 1$, the transformer could not approximate each $\boldsymbol{A}_{:,k}$ by $w_{1,k}\mathsf{softmax}(\boldsymbol{V}^{\top}\boldsymbol{Q}_1\boldsymbol{v}_k)$ in general since the entries of the latter vector are either all positive or all negative. In addition, Proposition 1 indicates that when $H \geq N$, the weights of the transformer with a simple initialization method satisfy our desired property that is crucial to guarantee the fast linear convergence. However, (18) implies that we should not set $H$ to be too large, since larger $H$ yields slower convergence rate.

## 4 Conclusion

We analyze the training dynamics of a one-layer transformer with multi-head softmax attention trained by gradient descent to solve complex non-linear regression tasks using partially labeled prompts. In this setting, the labels contain Gaussian noise, and each prompt may include only a few examples, which are insufficient to determine the underlying template. Our work overcomes several restrictive assumptions made in previous studies and proves that the training loss converges linearly to its minimum value. Furthermore, we analyze the transformer's strategy for addressing the issue of underdetermination during inference and evaluate its performance by comparing it with the best possible strategy. Our study provides the first analysis of how transformers can acquire contextual (template) information to generalize to unseen examples when prompts contain a limited number of query-answer pairs.

## Acknowledgments and Disclosure of Funding

The work of T. Yang and Y. Chi is supported in part by the grants NSF CCF-2007911, DMS-2134080 and ONR N00014-19-1-2404. The work of Y. Liang was supported in part by the U.S. National Science Foundation under the grants ECCS-2113860, DMS-2134145 and CNS-2112471.

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

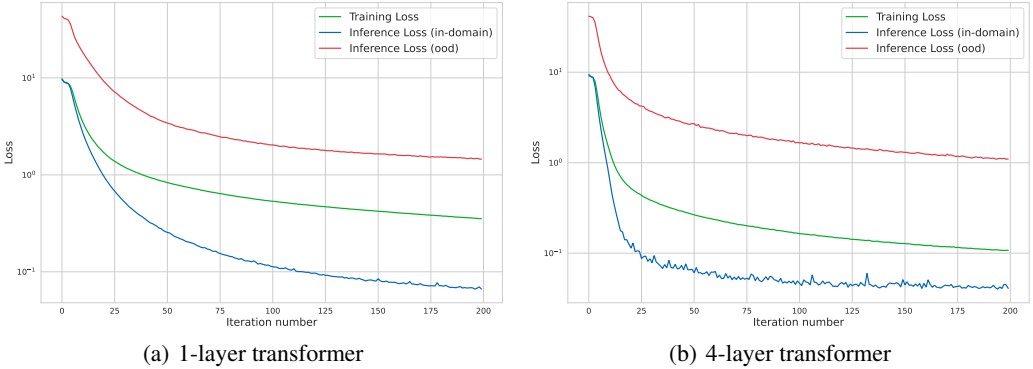

| (a) 1-layer transformer | (b) 4-layer transformer |

Figure 2: Training and inference losses of (a) 1-layer and (b) 4-layer transformers, which validate Theorem 2, as well as the transformer's contextual generalization to unseen examples and to unseen tasks.

## A    Experiments

This section aims to provide some empirical validation to our theoretical findings and verify that some of our results could be generalized to deeper transformers.

**Setup.**    We conduct experiments on a synthetic dataset, where we randomly generate each token $v_k$ and their representation $f(v_k)$ from standard Gaussian distribution. We employ both the 1-layer transformer described in Section 2 and a standard 4-layer transformer in Vaswani et al. [2017] with $d_{\text{model}} = 256$ and $d_{\text{ff}} = 512$. We set the training loss to be the population loss defined in (9), and initialize $\{Q_h^{(0)}\}_{h \in [H]}$ using standard Gaussian and set $\{w_h^{(0)}\}_{h \in [H]}$ to be $\mathbf{0}$, identical to what is specified in Section 3. We generate $\boldsymbol{\lambda}$ from standard Gaussian distribution to create the training set with 10000 samples and in-domain test set with 200 samples; we also create an out-of-domain (ood) test set with 200 samples by sampling $\boldsymbol{\lambda}$ from $\mathcal{N}(\mathbf{1}_m, 4\boldsymbol{I}_m)$. Given $\boldsymbol{\lambda}$, we generate the label $y_k$ of token $v_k$ using (1), for $k \in [K]$. We train with a batch size 256. All experiments use the Adam optimizer with a learning rate $1 \times 10^{-4}$.

**Training and inference performance.**    We set $N = 30$, $K = 200$, $d = 100$, $m = 20$, and set $H$ to be 64 and 8 for 1-layer and 4-layer transformers, respectively. Figure 2 shows the training and inference losses of both 1-layer and 4-layer transformers, where we measure the inference loss by $\frac{1}{K}\|\widehat{\boldsymbol{y}} - \widehat{\boldsymbol{y}}^\star\|_2^2$ to validate (22): after sufficient training, the output of the transformer $\widehat{\boldsymbol{y}}$ converges to $\widehat{\boldsymbol{y}}^\star$. From Figure 2 we can see that for both 1-layer and 4-layer transformers, the three curves have the same descending trend, despite the inference loss on the ood dataset is higher than that on the in-domain dataset. This experiment also shows the transformer's contextual generalization to unseen examples and to unseen tasks, validating our claim in Section 3.2.

Figure 3 plots the performance gap $\frac{1}{K}\left\|\widehat{\boldsymbol{y}}^\star - \widehat{\boldsymbol{y}}^{\text{best}}\right\|_2^2$ of the one-layer transformer with respect to different $N$ ranging from 50 to 150, when we fix $m = 100$ and $\tau = 0.01$. This verifies that the ridge regression implemented by the pretrained transformer has a better performance when $m$ is close to $N$, again verifying our claim at the end of Section 2.

**Impact of the number of attention heads.**    We now turn to examine the impact of the number of attention heads. In this experiment, we use the population loss (9), and set the other configurations same as those in Figure 2. Figure 4 shows the training loss curves for different $H$ with respect the iteration number, which validates our claims. From Figure 4, we can see that we need to set $H$ large enough to guarantee the convergence of the training loss. However, setting $H$ too large ($H = 400$) leads to instability and divergence of the loss. Recall that in Proposition 1, we require $H \geq N$ to guarantee our convergence results hold. Although this condition may not be necessary, Figure 4 shows that when $H < N = 30$, the loss stopped descending even when it is far from the minimal value. On the other side, the loss keeps descending when $H = 30$ (though slowly).

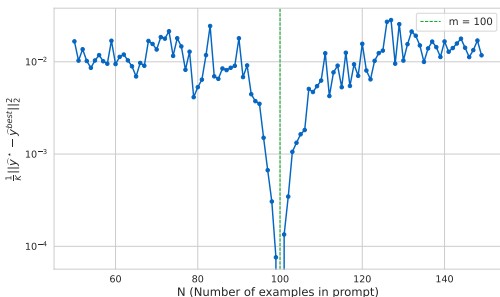

Figure 3: The performance gap $\frac{1}{K} \left\| \widehat{\boldsymbol{y}}^\star - \widehat{\boldsymbol{y}}^{\text{best}} \right\|_2^2$ with different $N$ when $m = 100$, which validates that the closer $N$ is to $m$, the better the transformer's prediction is.

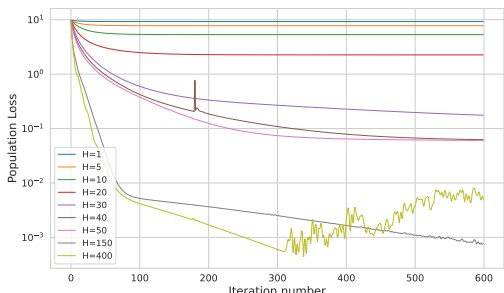

Figure 4: Training losses of the 1-layer transformer with different number of attention heads $H$, where $H$ should be large enough to guarantee the convergence of the training loss, but setting $H$ too large leads to instability and slower divergence.

We also explore how $H$ affects the training of the 4-layer transformer, as displayed in Figure 5, where we set $K = 200$ and the configurations other than $H$ are the same as in Figure 3. We fix the wall-clock time to be 100 seconds and plot the training loss curves with different $H$. Figure 5 (a) shows the final training and inference losses with respect to $H$. It reflects that the losses converge faster with smaller $H$ (here the final training loss is the smallest when $H = 4$). The training curves in Figure 5 (b) corresponding to different $H$ within 100s may provide some explanation to this phenomenon: (i) transformers with larger $H$ could complete less iterations within a fixed amount of time (the curves corresponding to larger $H$ are shorter); (ii) the training loss curves corresponding to large $H$ ($H = 32, 64$) descend more slowly. This suggests our claim that larger $H$ may yield slower convergence rate is still valid on deeper transformers. Note that unlike the 1-layer transformer, deeper transformers don't require a large $H$ to guarantee convergence. This is because deeper transformers have better expressive power even when $H$ is small.

# B  Proof Preparation

## B.1  Summary of key notation

We summarize the frequently used notation in Table 2 for ease of reference.

## B.2  Auxiliary lemmas

We provide some useful facts that will be repeatedly used later on. Let
$$\boldsymbol{z}_k := f(\boldsymbol{v}_k) = (f_1(\boldsymbol{v}_k), f_2(\boldsymbol{v}_k), \cdots, f_m(\boldsymbol{v}_k))^\top \in \mathbb{R}^m, \qquad \forall k \in [K].$$
Recalling (12), we can rewrite
$$\boldsymbol{Z} := (\boldsymbol{z}_1, \cdots, \boldsymbol{z}_N) \in \mathbb{R}^{m \times N}.$$
We further define $\boldsymbol{s}_k^h \in \mathbb{R}^N$ as follows:
$$\boldsymbol{s}_k^h := \mathsf{softmax}(\boldsymbol{V}^\top \boldsymbol{Q}_h \boldsymbol{v}_k) = (s_{1k}^h, \cdots, s_{Nk}^h)^\top, \quad \forall k \in [K], h \in [H]. \tag{28}$$

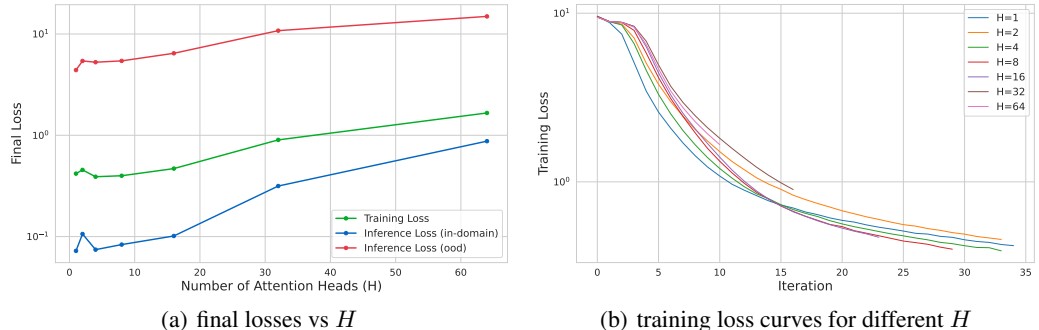

(a) final losses vs $H$          (b) training loss curves for different $H$

Figure 5: Training losses of a 4-layer transformer with different $H$, fixing wall-clock time to be 100s. This experiment shows that unlike 1-layer transformers, deeper transformers don't require $H$ to be large to guarantee convergence of the loss.

| notation | meaning |
|---|---|
| $K \in \mathbb{N}_+$ | total number of tokens |
| $d \in \mathbb{N}_+$ | token dimension |
| $m \in \mathbb{N}_+$ | number of basic tasks |
| $H \in \mathbb{N}_+$ | number of attention heads |
| $N \in \mathbb{N}_+$ | number of examples in each prompt |
| $\boldsymbol{v}_k \in \mathbb{R}^d, k \in [K]$ | the $k$-th token |
| $f_i : \mathbb{R}^d \to \mathbb{R}, i \in [m]$ | the $i$-th basic task |
| $\boldsymbol{\lambda} \in \mathbb{R}^m$ | coefficient vector |
| $y_k = \boldsymbol{\lambda}^\top (f(\boldsymbol{v}_k) + \boldsymbol{\epsilon}_k), k \in [K]$ | the $k$-th label |

Table 2: Notation for key parameters.

**Lemma 1** (Softmax gradient). *For all $j \in [N], k \in [K]$ and $h \in [H]$, we have*

$$\frac{\partial s_{jk}^h}{\partial \boldsymbol{Q}_h} = s_{jk}^h \sum_{i=1}^N s_{ik}^h (\boldsymbol{v}_j - \boldsymbol{v}_i) \boldsymbol{v}_k^\top, \tag{29}$$

*where $s_{jk}^h$ is defined in (28).*

*Proof.* See the proof of Lemma A.1 in Huang et al. [2023]. □

**Lemma 2** (Smoothness of softmax). *For vectors $\boldsymbol{\xi}_1, \boldsymbol{\xi}_2 \in \mathbb{R}^l$, we have*

$$\|\mathsf{softmax}(\boldsymbol{\xi}_1) - \mathsf{softmax}(\boldsymbol{\xi}_2)\|_1 \leq 2 \|\boldsymbol{\xi}_1 - \boldsymbol{\xi}_2\|_\infty. \tag{30}$$

*Proof.* See Corollary A.7 in Edelman et al. [2022]. □

We also need to make use of the following form of Young's inequality.

**Lemma 3.** *For any $\boldsymbol{x}_1, \cdots, \boldsymbol{x}_l \in \mathbb{R}^p$, we have*

$$\left\| \sum_{i=1}^l \boldsymbol{x}_i \right\|_2^2 \leq l \sum_{i=1}^l \|\boldsymbol{x}_i\|_2^2. \tag{31}$$

The following lemma shows the equivalence between (24) and (25).

**Lemma 4** (Equivalence of the regression problems). *Given any prompt $P_{\boldsymbol{\lambda}} := (\boldsymbol{v}_1, y_1, \cdots, \boldsymbol{v}_N, y_N)$, we have the following equivalence:*

$$\mathbb{E}_{\boldsymbol{\epsilon}} \left[ \frac{1}{2N} \sum_{i=1}^N (y_i - \boldsymbol{\lambda}^\top (f(\boldsymbol{v}_i) + \boldsymbol{\epsilon}_i))^2 \right] = \frac{1}{2N} \sum_{i=1}^N (y_i - \boldsymbol{\lambda}^\top f(\boldsymbol{v}_i))^2 + \frac{\tau}{2} \|\boldsymbol{\lambda}\|_2^2. \tag{32}$$

*Proof.* See Appendix E.2. □

## C    Proof of Theorem 1

We first outline the proof. To prove Theorem 1, we first remove the expectation in the expression of the loss function $\mathcal{L}$ in (9) by reformulating it to a deterministic form (see Lemma 5). With this new form, we show by induction that the loss function $\mathcal{L}$ is smooth (Lemma 10) and satisfies the Polyak-Łojasiewicz (PL) condition (c.f. (49)). Provided with both smoothness and PL conditions, we are able to give the desired linear convergence rate [Karimi et al., 2016].

We define

$$
\boldsymbol{\delta}_k^{\boldsymbol{\theta}} := \begin{cases} \sum_{h=1}^H w_{h,k} \boldsymbol{s}_k^h - \left( \boldsymbol{Z}^\top \boldsymbol{Z} + m\tau \boldsymbol{I} \right)^{-1} \left( \boldsymbol{z}_k + m\tau \boldsymbol{e}_k \right), & \text{if } k \in [N], \\ \sum_{h=1}^H w_{h,k} \boldsymbol{s}_k^h - \left( \boldsymbol{Z}^\top \boldsymbol{Z} + m\tau \boldsymbol{I} \right)^{-1} \boldsymbol{z}_k, & \text{if } k \in [K] \setminus [N]. \end{cases} \tag{33}
$$

We also define the following matrices:

$$
\boldsymbol{A} := \left( \boldsymbol{Z}^\top \boldsymbol{Z} + m\tau \boldsymbol{I}_N \right)^{-1} \left( \boldsymbol{Z}^\top \widehat{\boldsymbol{Z}} + (m\tau \boldsymbol{I}_N, \boldsymbol{0}) \right) \in \mathbb{R}^{N \times K}, \tag{34}
$$

$$
\widehat{\boldsymbol{A}}(\boldsymbol{\theta}) := \left( \sum_{h=1}^H w_{h,1} \boldsymbol{s}_1^h, \cdots, \sum_{h=1}^H w_{h,K} \boldsymbol{s}_K^h \right) \in \mathbb{R}^{N \times K}, \tag{35}
$$

where $\widehat{\boldsymbol{Z}} := (\boldsymbol{z}_1, \cdots, \boldsymbol{z}_K) \in \mathbb{R}^{m \times K}$.

We first reformulate the loss function to remove the expectation in the population loss.

**Lemma 5** (Reformulation of the loss function). *Under Assumption 1, the loss function $\mathcal{L}(\boldsymbol{\theta})$ could be rewritten into the following equivalent form:*

$$
\mathcal{L}(\boldsymbol{\theta}) = \frac{1}{2K} \sum_{k=1}^K \left\| \left( \boldsymbol{Z}^\top \boldsymbol{Z} + m\tau \boldsymbol{I} \right)^{1/2} \boldsymbol{\delta}_k^{\boldsymbol{\theta}} \right\|_2^2 + \mathcal{L}^\star = \frac{1}{2K} \sum_{k=1}^K \left\| \bar{\boldsymbol{Z}} \boldsymbol{\delta}_k^{\boldsymbol{\theta}} \right\|_2^2 + \mathcal{L}^\star, \tag{36}
$$

*where*

$$
\mathcal{L}^\star = \frac{1}{2K} \sum_{k=1}^N \left( - \left( \boldsymbol{Z}^\top \boldsymbol{z}_k + m\tau \boldsymbol{e}_k \right)^\top \left( \boldsymbol{Z}^\top \boldsymbol{Z} + m\tau \boldsymbol{I} \right)^{-1} \left( \boldsymbol{Z}^\top \boldsymbol{z}_k + m\tau \boldsymbol{e}_k \right) + \| \boldsymbol{z}_k \|_2^2 + m\tau \right)
$$

$$
+ \frac{1}{2K} \sum_{k=N+1}^K \left( - \left( \boldsymbol{Z}^\top \boldsymbol{z}_k \right)^\top \left( \boldsymbol{Z}^\top \boldsymbol{Z} + m\tau \boldsymbol{I} \right)^{-1} \left( \boldsymbol{Z}^\top \boldsymbol{z}_k \right) + \| \boldsymbol{z}_k \|_2^2 \right)
$$

*is a constant that does not depend on $\boldsymbol{\theta}$, and $\bar{\boldsymbol{Z}}$ is defined in (12).*

*Proof.* See Appendix E.3. □

Lemma 5 indicates that $\mathcal{L}^\star$ is a lower bound of $\mathcal{L}$. We'll later show that $\mathcal{L}^\star$ is actually the infimum of $\mathcal{L}$, i.e., $\mathcal{L}^\star = \inf_{\boldsymbol{\theta}} \mathcal{L}(\boldsymbol{\theta})$.

Lemma 5 also indicates that, the necessary and sufficient condition for $\mathcal{L}(\boldsymbol{\theta}^{(t)})$ to converge to $\mathcal{L}^\star$ during training is

$$
\forall k \in [K]: \quad \boldsymbol{\delta}_k^{\boldsymbol{\theta}^t} \to \boldsymbol{0}, \quad t \to \infty, \tag{37}
$$

which folllows immediately that (37) is equivalent to

$$
\widehat{\boldsymbol{A}}(\boldsymbol{\theta}^{(t)}) - \boldsymbol{A} \to \boldsymbol{0}, \quad t \to \infty. \tag{38}
$$

To simplify the analysis, we introduce the following reparameterization to unify the learning rates of all parameters, and we'll consider the losses after reparameterization in the subsequent proofs.

**Lemma 6** (Reparameterization). *Define*

$$
\gamma := \sqrt{\eta_w / \eta_Q}, \quad \boldsymbol{\alpha}_h := \boldsymbol{w}_h / \gamma, \quad \forall h \in [H], \tag{39}
$$

*and let*

$$\boldsymbol{\xi} := \{\boldsymbol{Q}_h, \boldsymbol{\alpha}_h\}_{h=1}^H, \quad \ell(\boldsymbol{\xi}) := \mathcal{L}(\boldsymbol{\theta}). \tag{40}$$

*Then* (10) *is equivalent to*

$$\boldsymbol{\xi}^{(t)} = \boldsymbol{\xi}^{(t-1)} - \eta_Q \nabla_{\boldsymbol{\xi}} \ell(\boldsymbol{\xi}^{(t-1)}), \quad \forall t \in [T]. \tag{41}$$

*Proof.* See Appendix E.4. □

We denote $\boldsymbol{\alpha}$ as $\boldsymbol{\alpha} := (\alpha_{h,k})_{h \in [H], k \in [K]} \in \mathbb{R}^{H \times K}$.

The following lemma bounds the gradient norms by the loss function, which is crucial to the proof of Theorem 1.

**Lemma 7** (Upper bound of the gradient norms). *Suppose Assumption 1 holds and* $|\alpha_{h,k}^{(t)}| \leq \alpha$. *Then for all* $h \in [H]$, *we have*

$$\left\| \frac{\partial \ell(\boldsymbol{\xi}^{(t)})}{\partial \boldsymbol{Q}_h^{(t)}} \right\|_F \leq 2\sqrt{2}\gamma \alpha \bar{f}_{\max} \sqrt{\ell(\boldsymbol{\xi}^{(t)}) - \mathcal{L}^\star}. \tag{42}$$

*Proof.* See Appendix E.5. □

Now we are ready to give the main proof.

*Proof of Theorem 1.* To prove Theorem 1, it suffices to prove that under our assumptions, we have

(Upper bound of the parameters:) $\quad \left\| \boldsymbol{\alpha}_h^{(t)} \right\|_2 \leq \alpha, \tag{43a}$

(Lower bound of eigenvalues:) $\quad \lambda_{\min} \left( \boldsymbol{B}_k^{(t)} \boldsymbol{B}_k^{(t)\top} \right) \geq \frac{\zeta_0}{2}, \tag{43b}$

(Linear decay of the loss:) $\quad \mathcal{L}(\boldsymbol{\theta}^{(t)}) - \mathcal{L}^\star \leq \left( 1 - \frac{\eta_Q \sigma}{2} \right)^t \left( \mathcal{L}(\boldsymbol{\theta}^{(0)}) - \mathcal{L}^\star \right), \tag{43c}$

where

$$\sigma := \frac{\zeta_0 \gamma^2}{K}, \quad \alpha := \sqrt{2K} \frac{4 \left\| \bar{\boldsymbol{Z}} \right\|_2}{\gamma \zeta_0} \sqrt{\mathcal{L}(\boldsymbol{\theta}^{(0)}) - \mathcal{L}^\star}, \tag{44}$$

and $\gamma, \boldsymbol{\alpha}_h$ is defined in (39), $\zeta_0$ is defined in (14). We shall prove (43a),(43b) and (43c) by induction.

**Base case.** It is apparent that (43a),(43b) and (43c) all hold when $t = 0$.

**Induction.** We make the following inductive hypothesis, i.e., when $s \in [t-1]$, (43a), (43b) and (43c) hold. Below we prove that (43a),(43b) and (43c) hold when $s = t$ by the following steps.

**Step 1: verify** (43b) **and the Polyak-Łojasiewicz condition.** We first compute the gradient of the loss w.r.t. $\boldsymbol{\alpha}$:

$$\forall k \in [K]: \quad \frac{\partial \ell(\boldsymbol{\xi})}{\partial \boldsymbol{\alpha}_k} = \frac{1}{2K} \frac{\partial}{\partial \boldsymbol{\alpha}_k} \left\| \bar{\boldsymbol{Z}} \boldsymbol{\delta}_k^\theta \right\|_2^2 = \frac{1}{2K} \frac{\partial}{\partial \boldsymbol{\alpha}_k} \left\| \bar{\boldsymbol{Z}} \left( \gamma \boldsymbol{C}_k \boldsymbol{\alpha}_k - \boldsymbol{A}_{:k} \right) \right\|_2^2$$
$$= \frac{\gamma}{K} \left( \bar{\boldsymbol{Z}} \boldsymbol{C}_k \right)^\top \bar{\boldsymbol{Z}} \boldsymbol{\delta}_k^\theta = \frac{\gamma}{K} \boldsymbol{B}_k^\top \bar{\boldsymbol{Z}} \boldsymbol{\delta}_k^\theta, \tag{45}$$

where the first equality follows from Lemma 5, $\boldsymbol{C}_k, \boldsymbol{B}_k$ is defined in (13).

Let $\boldsymbol{b}_k^h$ denote the $h$-th column vector of $\boldsymbol{B}_k$, $h \in [H]$, i.e., $\boldsymbol{B}_k := (\boldsymbol{b}_k^1, \cdots, \boldsymbol{b}_k^H)$, then for any $k \in [K]$ and $t \in \mathbb{N}_+$, we have

$$\left\| (\boldsymbol{b}_k^h)^{(t)} - (\boldsymbol{b}_k^h)^{(0)} \right\|_2 \leq \left\| \bar{\boldsymbol{Z}} \right\|_2 \left\| (\boldsymbol{s}_k^h)^{(t)} - (\boldsymbol{s}_k^h)^{(0)} \right\|_2$$
$$\leq \left\| \bar{\boldsymbol{Z}} \right\|_2 \left\| (\boldsymbol{s}_k^h)^{(t)} - (\boldsymbol{s}_k^h)^{(0)} \right\|_1$$
$$\leq 2 \left\| \bar{\boldsymbol{Z}} \right\|_2 \left\| \boldsymbol{V}^\top (\boldsymbol{Q}_h^{(t)} - \boldsymbol{Q}_h^{(0)}) \boldsymbol{v}_k \right\|_\infty$$
$$\leq 2 \left\| \bar{\boldsymbol{Z}} \right\|_2 \max_{j \in [N]} |\boldsymbol{v}_j^\top (\boldsymbol{Q}_h^{(t)} - \boldsymbol{Q}_h^{(0)}) \boldsymbol{v}_k|$$
$$\leq 2 \left\| \bar{\boldsymbol{Z}} \right\|_2 \left\| \boldsymbol{Q}_h^{(t)} - \boldsymbol{Q}_h^{(0)} \right\|_F, \tag{46}$$

where the third line uses Lemma 2, and that

$$\forall h \in [H]: \quad \left\|\boldsymbol{Q}_h^{(t)} - \boldsymbol{Q}_h^{(0)}\right\|_F \leq \sum_{s=0}^{t-1} \eta \left\|\frac{\partial \ell(\boldsymbol{\xi}^{(s)})}{\partial \boldsymbol{Q}_h^{(s)}}\right\|_F$$

$$\leq \sum_{s=0}^{t-1} 2\sqrt{2}\eta\gamma\alpha \bar{f}_{\max} \sqrt{\ell(\boldsymbol{\xi}^{(s)}) - \mathcal{L}^\star}$$

$$\leq 2\sqrt{2}\eta\gamma\alpha \bar{f}_{\max} \sqrt{\mathcal{L}(\boldsymbol{\theta}^{(0)}) - \mathcal{L}^\star} \sum_{s=0}^{t-1} \left(\sqrt{1 - \frac{\eta\sigma}{2}}\right)^s$$

$$\leq \frac{8\sqrt{2}\gamma\alpha \bar{f}_{\max}}{\sigma} \sqrt{\mathcal{L}(\boldsymbol{\theta}^{(0)}) - \mathcal{L}^\star}, \tag{47}$$

where the second inequality follows from Lemma 7 (cf. (42)) and the third inequality follows from the inductive hypothesis and the fact that $\ell(\boldsymbol{\xi}^{(s)}) = \mathcal{L}(\boldsymbol{\theta}^{(s)})$, $\forall s$. Combining (47) with (46), we have

$$\left\|\boldsymbol{B}_k^{(t)} - \boldsymbol{B}_k^{(0)}\right\|_F \leq 2\left\|\bar{\boldsymbol{Z}}\right\|_2 \sqrt{\sum_{h=1}^{H} \left\|\boldsymbol{Q}_h^{(t)} - \boldsymbol{Q}_h^{(0)}\right\|_F^2}$$

$$\leq \left\|\bar{\boldsymbol{Z}}\right\|_2 \sqrt{H} \frac{16\sqrt{2}\gamma\alpha \bar{f}_{\max}}{\sigma} \sqrt{\mathcal{L}(\boldsymbol{\theta}^{(0)}) - \mathcal{L}^\star}$$

$$\leq \left(1 - 1/\sqrt{2}\right) \sqrt{\zeta_0}, \tag{48}$$

where the last inequality follows from (15). The above inequality (48) indicates that

$$\forall \boldsymbol{x} \in \mathbb{R}^K: \quad \left\|\boldsymbol{x}^\top \boldsymbol{B}_k^{(t)}\right\|_2 \geq \left\|\boldsymbol{x}^\top \boldsymbol{B}_k^{(0)}\right\|_2 - \left\|\boldsymbol{x}^\top (\boldsymbol{B}_k^{(t)} - \boldsymbol{B}_k^{(0)})\right\|_2 \geq \sqrt{\zeta_0/2},$$

which gives (43b).

Therefore, we obtain the following PL condition:

$$\left\|\nabla_{\boldsymbol{\theta}} \ell(\boldsymbol{\xi}^{(t)})\right\|_F^2 \geq \sum_{k=1}^{K} \sum_{h=1}^{H} \left(\frac{\partial \ell(\boldsymbol{\xi})}{\partial \alpha_{h,k}}\right)^2 = \frac{\gamma^2}{K^2} \sum_{k=1}^{K} \left(\bar{\boldsymbol{Z}}\boldsymbol{\delta}_k^{(t)}\right)^\top \boldsymbol{B}_k^{(t)} \boldsymbol{B}_k^{(t)\top} \bar{\boldsymbol{Z}}\boldsymbol{\delta}_k^{(t)}$$

$$\geq \frac{\zeta_0 \gamma^2}{2K^2} \sum_{k=1}^{K} \left\|\bar{\boldsymbol{Z}}\boldsymbol{\delta}_k^{(t)}\right\|_2^2 = \sigma \left(\ell(\boldsymbol{\xi}^{(t)}) - \mathcal{L}^\star\right), \tag{49}$$

where the equality comes from (45), and the last equality follows from (36).

**Step 2: verify the smoothness of the loss function.** We first give the following lemma that bounds the Lipschitzness of $\boldsymbol{b}_k^h$ and $\boldsymbol{\delta}_k^\theta$, which will be used later on. For notation simplicity, we let $\boldsymbol{B}, \boldsymbol{Q}, \boldsymbol{\alpha}$ denote $\boldsymbol{B}(\boldsymbol{\theta}), \boldsymbol{Q}(\boldsymbol{\theta}), \boldsymbol{\alpha}(\boldsymbol{\theta})$, respectively, and let $\boldsymbol{B}', \boldsymbol{Q}', \boldsymbol{\alpha}'$ denote $\boldsymbol{B}(\boldsymbol{\theta}'), \boldsymbol{Q}(\boldsymbol{\theta}'), \boldsymbol{\alpha}(\boldsymbol{\theta}')$, respectively.

**Lemma 8** (Lipschitzness of $\boldsymbol{b}_k^h$ and $\boldsymbol{\delta}_k^\theta$). *For all $k \in [K]$ and $h \in [H]$, and all transformer parameters $\boldsymbol{\theta}, \boldsymbol{\theta}'$, if $\max\{|\alpha_{h,k}|, |\alpha'_{h,k}|\} \leq \alpha$, then we have*

$$\left\|\boldsymbol{b}_k^h(\boldsymbol{\theta}) - \boldsymbol{b}_k^h(\boldsymbol{\theta}')\right\|_2 \leq 2\left\|\bar{\boldsymbol{Z}}\right\|_2 \left\|\boldsymbol{Q}_h - \boldsymbol{Q}'_h\right\|_F, \tag{50}$$

$$\left\|\boldsymbol{\delta}_k^\theta - \boldsymbol{\delta}_k^{\theta'}\right\|_2 \leq 2\gamma\sqrt{H}\alpha \sqrt{\sum_{h=1}^{H} \left\|\boldsymbol{Q}_h - \boldsymbol{Q}'_h\right\|_F^2} + \gamma\sqrt{H} \left\|\boldsymbol{\alpha}_k - \boldsymbol{\alpha}'_k\right\|_2. \tag{51}$$

*Proof.* (50) follows from a similar argument in (46). Regarding the Lipschitzness of $\boldsymbol{\delta}_k^\theta$, we have

$$\left\|\boldsymbol{\delta}_k^\theta - \boldsymbol{\delta}_k^{\theta'}\right\|_2 = \gamma \left\|\sum_{h=1}^{H} \alpha_{h,k}(\boldsymbol{s}_k^h(\boldsymbol{\theta}) - \boldsymbol{s}_k^h(\boldsymbol{\theta}')) + \sum_{h=1}^{H} (\alpha_{h,k} - \alpha'_{h,k})\boldsymbol{s}_k^h(\boldsymbol{\theta}')\right\|_2$$

$$\leq \gamma \sum_{h=1}^{H} |\alpha_{h,k}| \left\|\boldsymbol{s}_k^h(\boldsymbol{\theta}) - \boldsymbol{s}_k^h(\boldsymbol{\theta}')\right\|_2 + \gamma \sum_{h=1}^{H} |\alpha_{h,k} - \alpha'_{h,k}| \left\|\boldsymbol{s}_k^h(\boldsymbol{\theta}')\right\|_2$$

$$\leq 2\gamma\sqrt{H}\alpha \sqrt{\sum_{h=1}^{H} \left\|\boldsymbol{Q}_h - \boldsymbol{Q}'_h\right\|_F^2} + \gamma\sqrt{H} \left\|\boldsymbol{\alpha}_k - \boldsymbol{\alpha}'_k\right\|_2,$$

where we use (46) again to bound the first term in the second line, and use the fact that $\left\|s_k^h(\boldsymbol{\theta}')\right\|_2 \leq 1$ and Cauchy-Schwarz inequality to bound the second term in the second line. $\qquad\square$

We also need the following lemma which bounds the norm of $\boldsymbol{B}_k$ and $\boldsymbol{\delta}_k^\theta$.

**Lemma 9** (Upper bounds of $\boldsymbol{b}_k^h$ and $\boldsymbol{\delta}_k^\theta$). *For all $k \in [K]$ and $h \in [H]$, if $\max\{|\alpha_{h,k}|, |\alpha_{h,k}'|\} \leq \alpha$, then we have*

$$\left\|\boldsymbol{b}_k^h\right\|_2 \leq \left\|\bar{\boldsymbol{Z}}\right\|_2, \tag{52}$$

$$\left\|\boldsymbol{\delta}_k^\theta\right\|_2 \leq \gamma H \alpha + \|\boldsymbol{A}\|_2, \tag{53}$$

*where $\boldsymbol{A}$ is defined in* (34).

*Proof.* (52) follows from

$$\left\|\boldsymbol{b}_k^h\right\|_2 \leq \left\|\bar{\boldsymbol{Z}}\right\|_2 \left\|s_k^h\right\|_2 \leq \left\|\bar{\boldsymbol{Z}}\right\|_2.$$

(53) follows from

$$\left\|\boldsymbol{\delta}_k^\theta\right\|_2 \leq \gamma \sum_{h=1}^{H} |\alpha_{h,k}| \left\|s_k^h\right\|_2 + \|\boldsymbol{A}\boldsymbol{e}_k\|_2 \leq \gamma H \alpha + \|\boldsymbol{A}\|_2.$$

$\qquad\square$

As a consequence of Lemma 8 and Lemma 9, For all $k \in [K]$, and all transformer parameters $\boldsymbol{\theta}, \boldsymbol{\theta}'$, if $\max\{|\alpha_{h,k}|, |\alpha_{h,k}'|\} \leq \alpha$, we have

$$
\begin{aligned}
&\|\nabla_{\boldsymbol{\alpha}_k}\ell(\boldsymbol{\xi}) - \nabla_{\boldsymbol{\alpha}_k}\ell(\boldsymbol{\xi}')\|_2 \\
&\overset{(45)}{=} \frac{\gamma}{K} \left\|(\boldsymbol{B}_k - \boldsymbol{B}_k')^\top \bar{\boldsymbol{Z}}\boldsymbol{\delta}_k^\theta + \boldsymbol{B}_k'^\top \bar{\boldsymbol{Z}}(\boldsymbol{\delta}_k^\theta - \boldsymbol{\delta}_k^{\theta'})\right\|_2 \\
&\leq \frac{\gamma}{K} \left\|\bar{\boldsymbol{Z}}\right\|_2 \|\boldsymbol{B}_k - \boldsymbol{B}_k'\|_F \left\|\boldsymbol{\delta}_k^\theta\right\|_2 + \frac{\gamma}{K} \left\|\bar{\boldsymbol{Z}}\right\|_2 \|\boldsymbol{B}_k'\|_F \left\|\boldsymbol{\delta}_k^\theta - \boldsymbol{\delta}_k^{\theta'}\right\|_2 \\
&\leq \frac{\gamma}{K} \cdot 2 \left\|\bar{\boldsymbol{Z}}\right\|_2^2 (2\gamma H\alpha + \|\boldsymbol{A}\|_2) \sqrt{\sum_{h=1}^{H} \|\boldsymbol{Q}_h - \boldsymbol{Q}_h'\|_F^2} + \frac{\gamma^2}{K} H \left\|\bar{\boldsymbol{Z}}\right\|_2^2 \|\boldsymbol{\alpha}_k - \boldsymbol{\alpha}_k'\|_2, \tag{54}
\end{aligned}
$$

from which we obtain the smoothness of the $\ell$ w.r.t. $\boldsymbol{\alpha}$ as follows:

$$
\begin{aligned}
&\|\nabla_{\boldsymbol{\alpha}}\ell(\boldsymbol{\xi}) - \nabla_{\boldsymbol{\alpha}}\ell(\boldsymbol{\xi}')\|_F^2 \\
&= \sum_{k=1}^{K} \|\nabla_{\boldsymbol{\alpha}_k}\ell(\boldsymbol{\xi}) - \nabla_{\boldsymbol{\alpha}_k}\ell(\boldsymbol{\xi}')\|_2^2 \\
&\leq 2K \left(\frac{\gamma}{K} \cdot 2 \left\|\bar{\boldsymbol{Z}}\right\|_2^2 (2\gamma H\alpha + \|\boldsymbol{A}\|_2)\right)^2 \sum_{h=1}^{H} \|\boldsymbol{Q}_h - \boldsymbol{Q}_h'\|_F^2 + 2\frac{\gamma^4}{K^2} H^2 \left\|\bar{\boldsymbol{Z}}\right\|_2^4 \|\boldsymbol{\alpha} - \boldsymbol{\alpha}'\|_F^2 \\
&\leq 2 \left(\frac{1}{K} \left(2\gamma \left\|\bar{\boldsymbol{Z}}\right\|_2^2 (2\gamma H\alpha + \|\boldsymbol{A}\|_2)\right)^2 + \frac{\gamma^4}{K^2} H^2 \left\|\bar{\boldsymbol{Z}}\right\|_2^4\right) \|\boldsymbol{\xi} - \boldsymbol{\xi}'\|_2^2, \tag{55}
\end{aligned}
$$

where the first inequality uses Young's inequality (c.f. Lemma 3).

To obtain the smoothness of the loss function w.r.t. $\boldsymbol{Q}_h$, we first note that by (82) we have

$$\frac{\partial \ell(\boldsymbol{\xi})}{\partial \boldsymbol{Q}_h} = \frac{\gamma}{K} \sum_{k=1}^{K} \sum_{j=1}^{N} \left(\bar{\boldsymbol{Z}}\boldsymbol{\delta}_k^\theta\right)^\top \boldsymbol{z}_j \cdot \alpha_{h,k} s_{jk}^h \sum_{i=1}^{N} s_{ik}^h (\boldsymbol{v}_j - \boldsymbol{v}_i)\boldsymbol{v}_k^\top. \tag{56}$$

Therefore, if $\max\{|\alpha_{h,k}|, |\alpha'_{h,k}|\} \le \alpha$, we have

$$
\begin{aligned}
\left\| \frac{\partial \ell(\boldsymbol{\xi})}{\partial \boldsymbol{Q}_h} - \frac{\partial \ell(\boldsymbol{\xi}')}{\partial \boldsymbol{Q}_h} \right\|_F &\le \frac{2\gamma \bar{f}_{\max}}{K} \sum_{k=1}^{K} \left\{ \sum_{j=1}^{N} \|\bar{\boldsymbol{Z}}\|_2 \left\| \boldsymbol{\delta}_k^{\theta} - \boldsymbol{\delta}_k^{\theta'} \right\|_2 \cdot \alpha s_{jk}^h(\boldsymbol{\theta}) \sum_{i=1}^{N} s_{ik}^h(\boldsymbol{\theta}) \right. \\
&\qquad + \sum_{j=1}^{N} \|\bar{\boldsymbol{Z}}\|_2 \left\| \boldsymbol{\delta}_k^{\theta'} \right\|_2 |\alpha_{h,k} - \alpha'_{h,k}| s_{jk}^h(\boldsymbol{\theta}) \sum_{i=1}^{N} s_{ik}^h(\boldsymbol{\theta}) \\
&\qquad + \sum_{j=1}^{N} \|\bar{\boldsymbol{Z}}\|_2 \left\| \boldsymbol{\delta}_k^{\theta'} \right\|_2 \alpha |s_{jk}^h(\boldsymbol{\theta}) - s_{jk}^h(\boldsymbol{\theta}')| \sum_{i=1}^{N} s_{ik}^h(\boldsymbol{\theta}) \\
&\qquad \left. + \sum_{j=1}^{N} \|\bar{\boldsymbol{Z}}\|_2 \left\| \boldsymbol{\delta}_k^{\theta'} \right\|_2 \alpha s_{jk}^h(\boldsymbol{\theta}') \sum_{i=1}^{N} |s_{ik}^h(\boldsymbol{\theta}) - s_{ik}^h(\boldsymbol{\theta}')| \right\} \\
&\le \frac{2\gamma \bar{f}_{\max} \|\bar{\boldsymbol{Z}}\|_2}{K} \sum_{k=1}^{K} \left\{ \left\| \boldsymbol{\delta}_k^{\theta} - \boldsymbol{\delta}_k^{\theta'} \right\|_2 \alpha + \left\| \boldsymbol{\delta}_k^{\theta'} \right\|_2 |\alpha_{h,k} - \alpha'_{h,k}| \right. \\
&\qquad \left. + \left\| \boldsymbol{\delta}_k^{\theta'} \right\|_2 \alpha \sum_{j=1}^{N} |s_{jk}^h(\boldsymbol{\theta}) - s_{jk}^h(\boldsymbol{\theta}')| + \left\| \boldsymbol{\delta}_k^{\theta'} \right\|_2 \alpha \sum_{i=1}^{N} |s_{ik}^h(\boldsymbol{\theta}) - s_{ik}^h(\boldsymbol{\theta}')| \right\} \\
&\le \frac{2\gamma \bar{f}_{\max} \|\bar{\boldsymbol{Z}}\|_2}{K} \sum_{k=1}^{K} \left\{ \left\| \boldsymbol{\delta}_k^{\theta} - \boldsymbol{\delta}_k^{\theta'} \right\|_2 \alpha + \left\| \boldsymbol{\delta}_k^{\theta'} \right\|_2 |\alpha_{h,k} - \alpha'_{h,k}| \right. \\
&\qquad \left. + 2 \left\| \boldsymbol{\delta}_k^{\theta'} \right\|_2 \alpha \sqrt{N} \left\| \boldsymbol{s}_k^h(\boldsymbol{\theta}) - \boldsymbol{s}_k^h(\boldsymbol{\theta}') \right\|_2 \right\},
\end{aligned}
\tag{57}
$$

where the third inequality uses Cauchy-Schwarz inequality. Combining the above inequality (57) with Lemma 8 and Lemma 9, we have

$$
\begin{aligned}
\left\| \frac{\partial \ell(\boldsymbol{\xi})}{\partial \boldsymbol{Q}_h} - \frac{\partial \ell(\boldsymbol{\xi}')}{\partial \boldsymbol{Q}_h} \right\|_F &\le \frac{2\gamma \bar{f}_{\max} \|\bar{\boldsymbol{Z}}\|_2}{K} \left\{ \alpha \gamma \sqrt{H} \left( 2K\alpha \sqrt{\sum_{h=1}^{H} \|\boldsymbol{Q}_h - \boldsymbol{Q}'_h\|_F^2} + \sqrt{K} \|\boldsymbol{\alpha} - \boldsymbol{\alpha}'\|_F \right) \right. \\
&\qquad + (\gamma H \alpha + \|\boldsymbol{A}\|_2) \sqrt{K} \|\boldsymbol{\alpha}_{h,:} - \boldsymbol{\alpha}'_{h,:}\|_2 \\
&\qquad \left. + (\gamma H \alpha + \|\boldsymbol{A}\|_2) \cdot 2\alpha \sqrt{N} \cdot 2K \|\boldsymbol{Q}'_h - \boldsymbol{Q}_h\|_F \right\},
\end{aligned}
\tag{58}
$$

where the last line uses (46) to bound $\left\| \boldsymbol{s}_k^h(\boldsymbol{\theta}) - \boldsymbol{s}_k^h(\boldsymbol{\theta}') \right\|_2$. The above inequality (58) further gives

$$
\begin{aligned}
&\sum_{h=1}^{H} \|\nabla_{\boldsymbol{Q}_h} \ell(\boldsymbol{\xi}) - \nabla_{\boldsymbol{Q}_h} \ell(\boldsymbol{\xi}')\|_F^2 \\
&\le 8 \cdot \frac{\gamma \bar{f}_{\max} \|\bar{\boldsymbol{Z}}\|_2}{K} \left\{ (2K\alpha)^2 \left[ (\alpha \gamma H)^2 + 4N (\alpha \gamma H + \|\boldsymbol{A}\|_2)^2 \right] \sum_{h=1}^{H} \|\boldsymbol{Q}_h - \boldsymbol{Q}'_h\|_F^2 \right. \\
&\qquad \left. + K \left[ (\alpha \gamma H)^2 + (\alpha \gamma H + \|\boldsymbol{A}\|_2)^2 \right] \|\boldsymbol{\alpha} - \boldsymbol{\alpha}'\|_F^2 \right\} \\
&\le 8\gamma \bar{f}_{\max} \|\bar{\boldsymbol{Z}}\|_2 \cdot \max\left\{1, (2\sqrt{K}\alpha)^2\right\} \left[ (\alpha \gamma H)^2 + 4N (\alpha \gamma H + \|\boldsymbol{A}\|_2)^2 \right] \|\boldsymbol{\xi}' - \boldsymbol{\xi}\|_2^2,
\end{aligned}
\tag{59}
$$

where the first inequality makes use of Young's inequality (c.f. Lemma 3).

Combining the above two relations (55) and (59), we obtain the smoothness of $\ell$ w.r.t. $\boldsymbol{\xi}$ as follows:

**Lemma 10** (Smoothness of the loss function). *Let $\gamma := \sqrt{\eta_w/\eta_Q}$. For all transformer parameters $\boldsymbol{\xi}, \boldsymbol{\xi}'$, if $\max\{|\alpha_{h,k}|, |\alpha'_{h,k}|\} \le \alpha$, then we have*

$$
\|\nabla_{\boldsymbol{\xi}} \ell(\boldsymbol{\xi}) - \nabla_{\boldsymbol{\xi}} \ell(\boldsymbol{\xi}')\|_2 \le L \|\boldsymbol{\xi} - \boldsymbol{\xi}'\|_2,
\tag{60}
$$

*where*

$$
\begin{aligned}
L^2 = 2 &\left( \frac{1}{K} \left( 2\gamma \|\bar{\boldsymbol{Z}}\|_2^2 (2\gamma H \alpha + \|\boldsymbol{A}\|_2) \right)^2 + \frac{\gamma^4}{K^2} H^2 \|\bar{\boldsymbol{Z}}\|_2^4 \right) \\
&+ 8\gamma \bar{f}_{\max} \|\bar{\boldsymbol{Z}}\|_2 \cdot \max\left\{1, (2\sqrt{K}\alpha)^2\right\} \left[ (\alpha \gamma H)^2 + 4N (\alpha \gamma H + \|\boldsymbol{A}\|_2)^2 \right].
\end{aligned}
\tag{61}
$$

**Step 3: verify** (43a). (45) implies

$$\frac{\partial \ell(\boldsymbol{\xi})}{\partial \alpha_{h,k}} = \frac{\gamma}{K} (\boldsymbol{b}_k^h)^\top \bar{\boldsymbol{Z}} \boldsymbol{\delta}_k^\theta,$$

which, combining with (52), gives

$$\forall k \in [K], h \in [H]: \quad \left( \frac{\partial \ell(\boldsymbol{\xi})}{\partial \alpha_{h,k}} \right)^2 \leq \frac{\gamma^2}{K^2} \left\| \bar{\boldsymbol{Z}} \right\|_2^2 \left\| \bar{\boldsymbol{Z}} \boldsymbol{\delta}_k^\theta \right\|_2^2.$$

Combining this with (36) we obtain

$$\left\| \frac{\ell(\boldsymbol{\xi})}{\partial \boldsymbol{\alpha}_h} \right\|_2^2 \leq \left\| \bar{\boldsymbol{Z}} \right\|_2^2 \frac{2\gamma^2}{K} \left( \ell(\boldsymbol{\xi}) - \mathcal{L}^\star \right),$$

which indicates

$$\left\| \frac{\partial \ell(\boldsymbol{\xi})}{\partial \boldsymbol{\alpha}_h} \right\|_2 \leq \left\| \bar{\boldsymbol{Z}} \right\|_2 \gamma \sqrt{\frac{2}{K} \left( \ell(\boldsymbol{\xi}) - \mathcal{L}^\star \right)}. \tag{62}$$

Therefore, we have

$$\begin{aligned}
\left\| \boldsymbol{\alpha}_h^{(t)} \right\|_2 &= \left\| \boldsymbol{\alpha}_h^{(0)} - \eta_Q \sum_{i=0}^{t-1} \frac{\partial \ell(\boldsymbol{\xi}^{(i)})}{\partial \boldsymbol{\alpha}_h} \right\|_2 \\
&\leq \left\| \boldsymbol{\alpha}_h^{(0)} \right\|_2 + \eta_Q \sum_{i=0}^{t-1} \left\| \frac{\partial \ell(\boldsymbol{\xi}^{(i)})}{\partial \boldsymbol{\alpha}_h} \right\|_2 \\
&\leq \left\| \boldsymbol{\alpha}_h^{(0)} \right\|_2 + \eta_Q \left\| \bar{\boldsymbol{Z}} \right\|_2 \sqrt{\frac{2\gamma^2}{K}} \sum_{i=0}^{t-1} \sqrt{\ell(\boldsymbol{\xi}^{(i)}) - \mathcal{L}^\star} \\
&\leq \left\| \boldsymbol{\alpha}_h^{(0)} \right\|_2 + \eta_Q \left\| \bar{\boldsymbol{Z}} \right\|_2 \sqrt{\frac{2\gamma^2 \left( \mathcal{L}(\boldsymbol{\theta}^{(0)}) - \mathcal{L}^\star \right)}{K}} \sum_{i=0}^{t-1} \left( \sqrt{1 - \frac{\eta_Q \sigma}{2}} \right)^i \\
&\leq \left\| \boldsymbol{\alpha}_h^{(0)} \right\|_2 + \eta_Q \left\| \bar{\boldsymbol{Z}} \right\|_2 \sqrt{\frac{2\gamma^2 \left( \mathcal{L}(\boldsymbol{\theta}^{(0)}) - \mathcal{L}^\star \right)}{K}} \cdot \frac{4}{\eta_Q \sigma},
\end{aligned}$$

where the second inequality follows from (62) and the third inequality follows from the induction hypothesis (43c). (43a) follows from plugging $\sigma$ defined in (44) into the above inequality and using the initializtion condition that $\boldsymbol{\alpha}^{(0)} = \frac{1}{\gamma} \boldsymbol{w}^{(0)} = \boldsymbol{0}$.

**Step 4: verify the linear convergence rate** (43c). Combining (43a), (60) and Lemma 4.3 in Nguyen and Mondelli [2020], we have

$$\ell(\boldsymbol{\xi}^{(t)}) - \mathcal{L}^\star \leq \ell(\boldsymbol{\xi}^{(t-1)}) - \mathcal{L}^\star + \langle \nabla_{\boldsymbol{\xi}} \ell(\boldsymbol{\xi}^{(t-1)}), \boldsymbol{\xi}^{(t)} - \boldsymbol{\xi}^{(t-1)} \rangle + \frac{L}{2} \left\| \boldsymbol{\xi}^{(t)} - \boldsymbol{\xi}^{(t-1)} \right\|_2^2, \tag{63}$$

which indicates when $\eta_Q \leq 1/L$, we have

$$\ell(\boldsymbol{\xi}^{(t)}) - \mathcal{L}^\star \leq \ell(\boldsymbol{\xi}^{(t-1)}) - \mathcal{L}^\star - \frac{\eta_Q}{2} \left\| \nabla_{\boldsymbol{\xi}} \ell(\boldsymbol{\xi}^{(t-1)}) \right\|_F^2 \overset{(49)}{\leq} \left( 1 - \frac{\eta_Q \sigma}{2} \right) \left( \ell(\boldsymbol{\xi}^{(t-1)}) - \mathcal{L}^\star \right), \tag{64}$$

which, combined with the fact that $\mathcal{L}(\boldsymbol{\theta}^{(s)}) = \ell(\boldsymbol{\xi}^{(s)})$ for all $s$ (see Lemma 6), verifies (43c).

Note that (36) implies that $\mathcal{L}^\star \leq \mathcal{L}(\boldsymbol{\theta})$ holds for all $\boldsymbol{\theta}$. And from (43c) we know that $\mathcal{L}(\boldsymbol{\theta}^{(t)}) \to \mathcal{L}^\star$ as $t \to \infty$. Therefore, it follows that

$$\mathcal{L}^\star = \inf_{\boldsymbol{\theta}} \mathcal{L}(\boldsymbol{\theta}).$$

Consequently, (43c) is equivalent to (18). □

# D Proof of Theorem 2

By (43c) we know that $\mathcal{L}(\boldsymbol{\theta}^{(t)}) \to \mathcal{L}^\star$ as $t \to \infty$. Thus from (36) we know that (37) and (38) hold.

By Sherman-Morrison-Woodbury formula, we have

$$\left(m\tau \boldsymbol{I}_N + \boldsymbol{Z}^\top \boldsymbol{Z}\right)^{-1} = \frac{1}{m\tau}\boldsymbol{I}_N - \frac{1}{m\tau}\boldsymbol{Z}^\top \left(m\tau \boldsymbol{I}_m + \boldsymbol{Z}\boldsymbol{Z}^\top\right)^{-1}\boldsymbol{Z}. \tag{65}$$

Thus we have

$$
\begin{aligned}
\boldsymbol{A} &\overset{(34)}{=} \left(\boldsymbol{Z}^\top \boldsymbol{Z} + m\tau \boldsymbol{I}_N\right)^{-1}\left(\boldsymbol{Z}^\top \widehat{\boldsymbol{Z}} + (m\tau \boldsymbol{I}_N, \boldsymbol{0})\right) \\
&\overset{(65)}{=} \frac{1}{m\tau}\left(\boldsymbol{I}_N - \boldsymbol{Z}^\top \left(m\tau \boldsymbol{I}_m + \boldsymbol{Z}\boldsymbol{Z}^\top\right)^{-1}\boldsymbol{Z}\right)\left(\boldsymbol{Z}^\top \widehat{\boldsymbol{Z}} + (m\tau \boldsymbol{I}_N, \boldsymbol{0})\right) \\
&= \frac{1}{m\tau}\Big[\boldsymbol{Z}^\top \widetilde{\boldsymbol{Z}} + (m\tau \boldsymbol{I}_N, \boldsymbol{0}) - \boldsymbol{Z}^\top \left(m\tau \boldsymbol{I}_m + \boldsymbol{Z}\boldsymbol{Z}^\top\right)^{-1}\left(m\tau \boldsymbol{I}_m + \boldsymbol{Z}\boldsymbol{Z}^\top\right)\widetilde{\boldsymbol{Z}} \\
&\qquad\quad + m\tau \boldsymbol{Z}^\top \left(m\tau \boldsymbol{I}_m + \boldsymbol{Z}\boldsymbol{Z}^\top\right)^{-1}\widetilde{\boldsymbol{Z}} - m\tau \boldsymbol{Z}^\top \left(m\tau \boldsymbol{I}_m + \boldsymbol{Z}\boldsymbol{Z}^\top\right)^{-1}(\boldsymbol{Z}, \boldsymbol{0})\Big] \\
&= (\boldsymbol{I}_N, \boldsymbol{0}) + \boldsymbol{Z}^\top \left(m\tau \boldsymbol{I}_m + \boldsymbol{Z}\boldsymbol{Z}^\top\right)^{-1}(\boldsymbol{0}, \boldsymbol{Z}^Q) \\
&= \left(\boldsymbol{I}_N, \boldsymbol{Z}^\top \left(m\tau \boldsymbol{I}_m + \boldsymbol{Z}\boldsymbol{Z}^\top\right)^{-1}\boldsymbol{Z}^Q\right),
\end{aligned} \tag{66}
$$

where $\boldsymbol{Z}^Q$ is defined in (19).

On the other hand, it's straightforward to verify that $\widehat{\boldsymbol{\lambda}}$ defined in (20) admits the following closed form:

$$\widehat{\boldsymbol{\lambda}} = \left(m\tau \boldsymbol{I}_m + \boldsymbol{Z}\boldsymbol{Z}^\top\right)^{-1}\boldsymbol{Z}\boldsymbol{y}. \tag{67}$$

Combining the above two equations, we obtain

$$\boldsymbol{A}^\top \boldsymbol{y} = \begin{pmatrix} \boldsymbol{y} \\ \left(\boldsymbol{Z}^Q\right)^\top \left(m\tau \boldsymbol{I}_m + \boldsymbol{Z}\boldsymbol{Z}^\top\right)^{-1}\boldsymbol{Z}\boldsymbol{y} \end{pmatrix} = \begin{pmatrix} \boldsymbol{y} \\ \left(\boldsymbol{Z}^Q\right)^\top \widehat{\boldsymbol{\lambda}} \end{pmatrix} = \widehat{\boldsymbol{y}}^\star,$$

where the last equality follows from (22).

Now we give the iteration complexity for the mean-squared error between the prediction $\widehat{\boldsymbol{y}}$ and the limit point $\widehat{\boldsymbol{y}}^\star$ to be less than $\varepsilon$. Given any prompt $P = P_{\boldsymbol{\lambda}}$, where $\boldsymbol{\lambda}$ satisfies Assumption 4, we have

$$y_i = \boldsymbol{\lambda}^\top (\boldsymbol{z}_i + \boldsymbol{\epsilon}_i) \sim \mathcal{N}(\boldsymbol{\lambda}^\top \boldsymbol{z}_i, \|\boldsymbol{\lambda}\|_2^2 \tau).$$

Letting $x_i = \frac{y_i - \boldsymbol{\lambda}^\top \boldsymbol{z}_i}{\|\boldsymbol{\lambda}\|_2 \sqrt{\tau}}$, we have $x_i \sim \mathcal{N}(0, 1)$. Define

$$Z = \sum_{i=1}^N \|\boldsymbol{\lambda}\|_2^2 \tau(x_i^2 - 1) = \left\|\boldsymbol{y} - \boldsymbol{Z}^\top \boldsymbol{\lambda}\right\|_2^2 - N\tau \|\boldsymbol{\lambda}\|_2^2.$$

By Laurent and Massart [2000, Lemma 1], we have

$$\forall s > 0: \quad \mathbb{P}\left(Z \geq 2\sqrt{N}\|\boldsymbol{\lambda}\|_2^2 \tau \sqrt{s} + 2\|\boldsymbol{\lambda}\|_2^2 \tau s\right) \leq \exp(-s).$$

By letting $s = \log(1/\delta)$ and using the definition of $Z$, we have

$$\mathbb{P}\left(\left\|\boldsymbol{y} - \boldsymbol{Z}^\top \boldsymbol{\lambda}\right\|_2^2 \geq N\tau \|\boldsymbol{\lambda}\|_2^2 + 2\sqrt{N\log(1/\delta)}\|\boldsymbol{\lambda}\|_2^2 \tau + 2\|\boldsymbol{\lambda}\|_2^2 \tau \log(1/\delta)\right) \leq \delta. \tag{68}$$

Thus with probability at least $1 - \delta$, we have

$$
\begin{aligned}
\|\boldsymbol{y}\|_2 &\leq \left\|\boldsymbol{Z}^\top \boldsymbol{\lambda}\right\|_2 + \left\|\boldsymbol{y} - \boldsymbol{Z}^\top \boldsymbol{\lambda}\right\|_2 \\
&\leq \left\|\boldsymbol{Z}^\top \boldsymbol{\lambda}\right\|_2 + \|\boldsymbol{\lambda}\|_2 \sqrt{\tau}\left(N + 2\sqrt{N\log(1/\delta)} + 2\log(1/\delta)\right)^{1/2} \\
&\leq B\left(\|\boldsymbol{Z}\|_2 + \sqrt{\tau}\left(N + 2\sqrt{N\log(1/\delta)} + 2\log(1/\delta)\right)^{1/2}\right).
\end{aligned} \tag{69}
$$

where we use (68) in the second inequality, and the third inequality follows from Assumption 4.

On the other hand, by (36) we have

$$\mathcal{L}(\boldsymbol{\theta}^{(t)}) = \frac{1}{2K} \left\| \bar{\boldsymbol{Z}}(\widehat{\boldsymbol{A}} - \boldsymbol{A}) \right\|_2^2 + \mathcal{L}^\star \geq \frac{m\tau}{2K} \left\| \widehat{\boldsymbol{A}} - \boldsymbol{A} \right\|_2^2 + \mathcal{L}^\star,$$

which gives

$$\left\| \widehat{\boldsymbol{A}} - \boldsymbol{A} \right\|_2 \leq \sqrt{\frac{2K}{m\tau} \left( \mathcal{L}(\boldsymbol{\theta}^{(T)}) - \mathcal{L}^\star \right)} \leq \sqrt{\frac{2K}{m\tau} \left( \mathcal{L}(\boldsymbol{\theta}^{(0)}) - \mathcal{L}^\star \right)} \left( 1 - \frac{\gamma^2 \eta_Q \zeta_0}{2K} \right)^{T/2}. \qquad (70)$$

Thus we know that w.p. at least $1 - \delta$, we have

$$\frac{1}{2K} \|\widehat{\boldsymbol{y}} - \widehat{\boldsymbol{y}}^\star\|_2^2 = \frac{1}{2K} \left\| \left( \widehat{\boldsymbol{A}} - \boldsymbol{A} \right)^\top \boldsymbol{y} \right\|_2^2 \leq \frac{1}{2K} \left\| \widehat{\boldsymbol{A}} - \boldsymbol{A} \right\|_2^2 \|\boldsymbol{y}\|_2^2 \leq \varepsilon,$$

where the last relation follows from (69), (70) and (21).

# E    Proof of Key Lemmas

## E.1    Proof of Proposition 1

For notation simplicity we drop the superscript $(0)$ in the subsequent proof.

Let $\boldsymbol{D}_k := \left( \boldsymbol{V}^\top \boldsymbol{Q}_1 \boldsymbol{v}_k, \cdots, \boldsymbol{V}^\top \boldsymbol{Q}_H \boldsymbol{v}_k \right) \in \mathbb{R}^{N \times H}$. Note that

$$\boldsymbol{D}_k = \boldsymbol{V}^\top \boldsymbol{Q} = \boldsymbol{V}^\top (\boldsymbol{q}_1, \cdots, \boldsymbol{q}_H), \quad \text{where } \boldsymbol{Q}(i,j) \overset{i.i.d.}{\sim} \mathcal{N}(0, \beta^2 \|\boldsymbol{v}_k\|_2^2), \quad \forall i \in [d], j \in [H]. \tag{71}$$

This suggests the column vectors of $\boldsymbol{D}_k$ are i.i.d. and the density of each column vector is positive at any point $\boldsymbol{x} \in \mathcal{R}(\boldsymbol{V})$, where $\mathcal{R}(\boldsymbol{V}) \subset \mathbb{R}^N$ is the row space of $\boldsymbol{V}$.

Since $\bar{\boldsymbol{Z}}$ has full rank, to prove $\boldsymbol{B}_k$ has full rank a.s., we only need to argue that $\boldsymbol{C}_k(:, 1 : N)$ has full rank w.p. 1. Below we prove this by contradiction (recall that by definition $\boldsymbol{C}_k = \mathsf{softmax}(\boldsymbol{D}_k)$, and we assume $H \geq N$).

Suppose w.p. larger than 0, there exists one of $\boldsymbol{C}_k(:, 1 : N)$'s column vector that could be linearly represented by its other $N - 1$ column vectors. Without loss of generality, we assume this colomn vector is $\boldsymbol{C}_k(:, 1) = \mathsf{softmax}(\boldsymbol{D}_k(:, 1))$. Let $\boldsymbol{x} = \boldsymbol{x}(\boldsymbol{q}_1) := \exp(\boldsymbol{D}_k(:, 1)) = \exp(\boldsymbol{V}^\top \boldsymbol{q}_1)$. Then $\boldsymbol{x}$ could be linearly represented by $\exp(\boldsymbol{D}_k(:, i))$, $i = 2, \cdots, N$.

Let $\tilde{\boldsymbol{A}} := \exp(\boldsymbol{D}_k(:, 2 : N))$, then w.p. larger than 0, $\boldsymbol{x} \in \mathcal{C}(\tilde{\boldsymbol{A}})$, where $\mathcal{C}(\tilde{\boldsymbol{A}})$ is the column vector space of $\tilde{\boldsymbol{A}}$. i.e., we have

$$\int_{\mathbb{R}^{N \times (m-1)}} \mathbb{P}(\boldsymbol{x} \in \mathcal{C}(\tilde{\boldsymbol{A}}) | \tilde{\boldsymbol{A}}) d\mu(\tilde{\boldsymbol{A}}) > 0,$$

which further indicates that there exists $\tilde{\boldsymbol{A}} \in \mathbb{R}^{N \times (N-1)}$ such that $\mathbb{P}(\boldsymbol{x} \in \mathcal{C}(\tilde{\boldsymbol{A}})) > 0$. Since the dimension of $\mathcal{C}(\tilde{\boldsymbol{A}})$ is at most $N - 1$, there exists $\boldsymbol{y} \in \mathbb{R}^N$, $\boldsymbol{y} \neq \boldsymbol{0}$ such that $\boldsymbol{y} \perp \mathcal{C}(\tilde{\boldsymbol{A}})$. Therefore, we have

$$\mathbb{P}(\boldsymbol{y}^\top \boldsymbol{x} = 0) > 0. \tag{72}$$

By Assumption 3, without loss of generality, we assume that $\boldsymbol{u}_1 = (v_{11}, v_{12}, \cdots, v_{1N})^\top$ has different entries. For any vector $\boldsymbol{w} = (w_1, \cdots, w_d)^\top \in \mathbb{R}^d$, we let $\tilde{\boldsymbol{w}} = (w_2, \cdots, w_d)^\top \in \mathbb{R}^{d-1}$ denote the vector formed by deleting the first entry of $\boldsymbol{w}$. Let $\boldsymbol{q}_1 = (q, \tilde{\boldsymbol{q}}_1^\top)^\top$. For any fixed $\tilde{\boldsymbol{q}}_1 \in \mathbb{R}^{d-1}$, the function $g(\cdot | \tilde{\boldsymbol{q}}_1) : \mathbb{R} \to \mathbb{R}$ defined by

$$g(q | \tilde{\boldsymbol{q}}_1) := \sum_{i=1}^N y_i e^{q v_{1i} + \tilde{\boldsymbol{q}}_1^\top \tilde{\boldsymbol{v}}_i} = \sum_{i=1}^N y_i e^{\tilde{\boldsymbol{q}}_1^\top \tilde{\boldsymbol{v}}_i} e^{q v_{1i}} = \left\langle \boldsymbol{y}, \exp(\boldsymbol{V}^\top \boldsymbol{q}_1) \right\rangle = \left\langle \boldsymbol{y}, \boldsymbol{x}(\boldsymbol{q}_1) \right\rangle$$

has finite zero points and thus $\{q \in \mathbb{R} | g(q | \tilde{\boldsymbol{q}}_1) = 0\}$ is a zero-measure set. Therefore, we have

$$\mathbb{P}(\langle \boldsymbol{y}, \boldsymbol{x} \rangle = 0) = \int_{\mathbb{R}^{d-1}} \mathbb{P}(g(q | \tilde{\boldsymbol{q}}_1) = 0 | \tilde{\boldsymbol{q}}_1) d\mu(\tilde{\boldsymbol{q}}_1) = 0,$$

which contradicts (72). Therefore, $\boldsymbol{C}_k(:, 1 : N)$ has full rank with probability 1.

## E.2 Proof of Lemma 4

Lemma 4 can be verified by the following direct computation (recall that the noise in each label satisfies $\boldsymbol{\epsilon}_i \overset{i.i.d}{\sim} \mathcal{N}(0, \tau \boldsymbol{I}_m), \forall i \in [N]$):

$$
\begin{aligned}
\mathbb{E}_{\boldsymbol{\epsilon}} & \left[ \frac{1}{2N} \sum_{i=1}^N (y_i - \boldsymbol{\lambda}^\top (f(\boldsymbol{v}_i) + \boldsymbol{\epsilon}_i))^2 \right] \\
&= \mathbb{E}_{\boldsymbol{\epsilon}} \left[ \frac{1}{2N} \sum_{i=1}^N \left( (y_i - \boldsymbol{\lambda}^\top f(\boldsymbol{v}_i))^2 - 2\boldsymbol{\lambda}^\top \boldsymbol{\epsilon}_i (y_i - \boldsymbol{\lambda}^\top f(\boldsymbol{v}_i)) + \boldsymbol{\lambda}^\top \boldsymbol{\epsilon}_i \boldsymbol{\epsilon}_i^\top \boldsymbol{\lambda} \right) \right] \\
&= \frac{1}{2N} \sum_{i=1}^N \left( (y_i - \boldsymbol{\lambda}^\top f(\boldsymbol{v}_i))^2 + \tau \|\boldsymbol{\lambda}\|_2^2 \right) \\
&= \frac{1}{2N} \sum_{i=1}^N (y_i - \boldsymbol{\lambda}^\top f(\boldsymbol{v}_i))^2 + \frac{\tau}{2} \|\boldsymbol{\lambda}\|_2^2 .
\end{aligned}
$$

## E.3 Proof of Lemma 5

We let $\boldsymbol{\epsilon}^P := (\boldsymbol{\epsilon}_1, \cdots, \boldsymbol{\epsilon}_N) \in \mathbb{R}^{m \times N}$, $\boldsymbol{\epsilon} := (\boldsymbol{\epsilon}_1, \cdots, \boldsymbol{\epsilon}_K) \in \mathbb{R}^{m \times K}$. Recall that $\boldsymbol{y} = (y_1, \cdots, y_N)^\top \in \mathbb{R}^N$. Then we have

$$
\boldsymbol{y} = (\boldsymbol{Z} + \boldsymbol{\epsilon}^P)^\top \boldsymbol{\lambda}, \tag{73}
$$

and

$$
\mathcal{L}(\boldsymbol{\theta}) = \frac{1}{K} \sum_{k=1}^K \mathcal{L}_k(\boldsymbol{\theta}) = \frac{1}{2} \mathbb{E}_{\boldsymbol{\lambda},\boldsymbol{\epsilon}} \left[ \frac{1}{K} \sum_{k=1}^K (\widehat{y}_k - y_k)^2 \right] \tag{74}
$$

$$
= \frac{1}{2K} \sum_{k=1}^K \mathbb{E}_{\boldsymbol{\lambda},\boldsymbol{\epsilon}} \left\| \boldsymbol{y}^\top \widehat{\boldsymbol{a}}_k - \boldsymbol{\lambda}^\top (\boldsymbol{z}_k + \boldsymbol{\epsilon}_k) \right\|_2^2
$$

$$
= \frac{1}{2K} \sum_{k=1}^K \mathbb{E}_{\boldsymbol{\lambda},\boldsymbol{\epsilon}} \left\| \boldsymbol{\lambda}^\top (\boldsymbol{Z} + \boldsymbol{\epsilon}^P) \widehat{\boldsymbol{a}}_k - \boldsymbol{\lambda}^\top (\boldsymbol{z}_k + \boldsymbol{\epsilon}_k) \right\|_2^2
$$

$$
= \frac{1}{2K} \sum_{k=1}^K \mathbb{E}_{\boldsymbol{\lambda},\boldsymbol{\epsilon}} \left[ (\boldsymbol{Z} + \boldsymbol{\epsilon}^P)\widehat{\boldsymbol{a}}_k - (\boldsymbol{z}_k + \boldsymbol{\epsilon}_k) \right]^\top \boldsymbol{\lambda}\boldsymbol{\lambda}^\top \left[ (\boldsymbol{Z} + \boldsymbol{\epsilon}^P)\widehat{\boldsymbol{a}}_k - (\boldsymbol{z}_k + \boldsymbol{\epsilon}_k) \right]
$$

$$
= \frac{1}{2K} \sum_{k=1}^K \mathbb{E}_{\boldsymbol{\epsilon}} \left[ (\boldsymbol{Z} + \boldsymbol{\epsilon}^P)\widehat{\boldsymbol{a}}_k - (\boldsymbol{z}_k + \boldsymbol{\epsilon}_k) \right]^\top \left[ (\boldsymbol{Z} + \boldsymbol{\epsilon}^P)\widehat{\boldsymbol{a}}_k - (\boldsymbol{z}_k + \boldsymbol{\epsilon}_k) \right]
$$

$$
= \frac{1}{2K} \sum_{k=1}^K \mathbb{E}_{\boldsymbol{\epsilon}} \left[ \|\boldsymbol{Z}\widehat{\boldsymbol{a}}_k - \boldsymbol{z}_k\|_2^2 + 2(\boldsymbol{Z}\widehat{\boldsymbol{a}}_k - \boldsymbol{z}_k)^\top (\boldsymbol{\epsilon}^P \widehat{\boldsymbol{a}}_k - \boldsymbol{\epsilon}_k) + \left\| \boldsymbol{\epsilon}^P \widehat{\boldsymbol{a}}_k - \boldsymbol{\epsilon}_k \right\|_2^2 \right], \tag{75}
$$

where $\widehat{\boldsymbol{a}}_k$ denote the $k$-th column vector of matrix $\widehat{\boldsymbol{A}}(\boldsymbol{\theta})$ defined in (35), and the fifth line uses Assumption 1.

Note that for all $k \in [K]$, we have

$$
\mathbb{E}_{\boldsymbol{\epsilon}}(\boldsymbol{Z}\widehat{\boldsymbol{a}}_k - \boldsymbol{z}_k)^\top (\boldsymbol{\epsilon}^P \widehat{\boldsymbol{a}}_k - \boldsymbol{\epsilon}_k) = 0, \tag{76}
$$

and that

$$
\mathbb{E}_{\boldsymbol{\epsilon}} \left\| \boldsymbol{\epsilon}^P \widehat{\boldsymbol{a}}_k - \boldsymbol{\epsilon}_k \right\|_2^2 = m\tau \left( \|\widehat{\boldsymbol{a}}_k\|_2^2 + 1 \right) - 2m\tau \widehat{a}_{kk} \mathbb{1}\{k \in [N]\}, \tag{77}
$$

where $\mathbb{1}\{k \in [N]\}$ is the indicator function that equals 1 if $k \in [N]$ and 0 otherwise, and we have made use of the assumption that $\boldsymbol{\epsilon}_k \overset{i.i.d.}{\sim} \mathcal{N}(0, \tau^2 \boldsymbol{I}_m)$.

Combining the above two equations with (75), we know that for $k \in [N]$, it holds that

$$
\mathcal{L}_k(\boldsymbol{\theta}) = \frac{1}{2} \left( \|\boldsymbol{Z}\widehat{\boldsymbol{a}}_k - \boldsymbol{z}_k\|_2^2 + m\tau \|\widehat{\boldsymbol{a}}_k - \boldsymbol{e}_k\|_2^2 \right).
$$

Reorganizing the terms in the RHS of the above equation, we obtain that

$$\mathcal{L}_k(\boldsymbol{\theta}) = \frac{1}{2} \left\| \left(\boldsymbol{Z}^\top \boldsymbol{Z} + m\tau\boldsymbol{I}\right)^{1/2} \left(\widehat{\boldsymbol{a}}_k - \left(\boldsymbol{Z}^\top \boldsymbol{Z} + m\tau\boldsymbol{I}\right)^{-1} \left(\boldsymbol{Z}^\top \boldsymbol{z}_k + m\tau\boldsymbol{e}_k\right)\right) \right\|_2^2 + \frac{1}{2} c_k, \quad (78)$$

where $c_k = -\left(\boldsymbol{Z}^\top \boldsymbol{z}_k + m\tau\boldsymbol{e}_k\right)^\top \left(\boldsymbol{Z}^\top \boldsymbol{Z} + m\tau\boldsymbol{I}\right)^{-1} \left(\boldsymbol{Z}^\top \boldsymbol{z}_k + m\tau\boldsymbol{e}_k\right) + \|\boldsymbol{z}_k\|_2^2 + m\tau$.

By a similar argument, we can show that for $k \in [K]\backslash[N]$, it holds thet

$$\mathcal{L}_k(\boldsymbol{\theta}) = \frac{1}{2} \left\| \left(\boldsymbol{Z}^\top \boldsymbol{Z} + m\tau\boldsymbol{I}\right)^{1/2} \left(\widehat{\boldsymbol{a}}_k - \left(\boldsymbol{Z}^\top \boldsymbol{Z} + m\tau\boldsymbol{I}\right)^{-1} \boldsymbol{Z}^\top \boldsymbol{z}_k\right) \right\|_2^2 + \frac{1}{2} c'_k, \quad (79)$$

where $c'_k = -\left(\boldsymbol{Z}^\top \boldsymbol{z}_k\right)^\top \left(\boldsymbol{Z}^\top \boldsymbol{Z} + m\tau\boldsymbol{I}\right)^{-1} \left(\boldsymbol{Z}^\top \boldsymbol{z}_k\right) + \|\boldsymbol{z}_k\|_2^2$.

(78), (79) together with (33) and the definition of $\mathcal{L}^\star$ give (36).

### E.4  Proof of Lemma 6

First, it holds that

$$\boldsymbol{Q}_h^{(t)} = \boldsymbol{Q}_h^{(t-1)} - \eta_Q \nabla_{\boldsymbol{Q}_h} \ell(\boldsymbol{\xi}^{(t-1)}) = \boldsymbol{Q}_h^{(t-1)} - \eta_Q \nabla_{\boldsymbol{Q}_h} \ell(\boldsymbol{\xi}^{(t-1)}). \quad (80)$$

Second, note that

$$\begin{aligned}
\boldsymbol{w}_h^{(t)} &= \boldsymbol{w}_h^{(t-1)} - \eta_w \nabla_{\boldsymbol{w}_h} \mathcal{L}(\boldsymbol{\theta}^{(t-1)}) \\
&= \gamma \boldsymbol{\alpha}_h^{(t-1)} - \gamma^2 \cdot \frac{1}{\gamma} \eta_Q \nabla_{\boldsymbol{\alpha}_h} \ell(\boldsymbol{\xi}^{(t-1)}) \\
&= \gamma \left(\boldsymbol{\alpha}_h^{(t-1)} - \eta_Q \nabla_{\boldsymbol{\alpha}_h} \ell(\boldsymbol{\xi}^{(t-1)})\right).
\end{aligned}$$

Dividing both sides of the above equality by $\gamma$, we have

$$\boldsymbol{\alpha}_h^{(t)} = \boldsymbol{\alpha}_h^{(t-1)} - \eta_Q \nabla_{\boldsymbol{\alpha}_h} \ell(\boldsymbol{\xi}^{(t-1)}). \quad (81)$$

Hence, (41) follows from combining (80) and (81).

### E.5  Proof of Lemma 7

Throughout this proof, we omit the superscript $(t)$ for simplicity. We first compute the gradient of $\mathcal{L}$ w.r.t. $\boldsymbol{Q}_h$. By (36) we know that

$$\ell(\boldsymbol{\xi}) = \mathcal{L}(\boldsymbol{\theta}) = \frac{1}{2K} \sum_{k=1}^K \left\| \bar{\boldsymbol{Z}} \boldsymbol{\delta}_k \right\|_2^2,$$

and thus we have

$$\begin{aligned}
\frac{\partial \ell(\boldsymbol{\xi})}{\partial \boldsymbol{Q}_h} &= \frac{1}{K} \sum_{k=1}^K \sum_{j=1}^N \frac{\partial}{\partial \delta_{jk}} \left[ \frac{1}{2} \left\| \sum_{i=1}^N \delta_{ik} \bar{\boldsymbol{z}}_i \right\|_2^2 \right] \frac{\partial \delta_{jk}}{\partial \boldsymbol{Q}_h} \\
&= \frac{\gamma}{K} \sum_{k=1}^K \sum_{j=1}^N \left( \bar{\boldsymbol{Z}} \boldsymbol{\delta}_k \right)^\top \bar{\boldsymbol{z}}_j \cdot \underbrace{\alpha_{h,k} s_{jk}^h \sum_{i=1}^N s_{ik}^h (\boldsymbol{v}_j - \boldsymbol{v}_i) \boldsymbol{v}_k^\top}_{=:\boldsymbol{G}^{h,jk}}.
\end{aligned} \quad (82)$$

Note that

$$\left\| \boldsymbol{G}^{h,jk} \right\|_F \le 2\alpha s_{jk}^h, \quad (83)$$

where we use the fact that $\left\|(v_j - v_i)v_k^\top\right\|_2 \leq 2$ (recall that we assume each $v_k$ has unit norm, $k \in [K]$.) Combining (82) and (83), we have the desired result

$$\left\|\frac{\partial\ell(\boldsymbol{\xi})}{\partial\boldsymbol{Q}_h}\right\|_F \leq \frac{\gamma}{K}\sum_{k=1}^K\sum_{j=1}^N\left\|\bar{\boldsymbol{Z}}\boldsymbol{\delta}_k\right\|_2\left\|\bar{\boldsymbol{z}}_j\right\|_2\left\|\boldsymbol{G}^{h,jk}\right\|_F$$

$$\leq \frac{2\gamma}{K}\sum_{k=1}^K\sum_{j=1}^N\left\|\bar{\boldsymbol{Z}}\boldsymbol{\delta}_k\right\|_2\bar{f}_{\max}\alpha s_{jk}^h$$

$$\leq \frac{2\gamma\bar{f}_{\max}\alpha}{K}\sqrt{K}\sqrt{\sum_{k=1}^K\left\|\bar{\boldsymbol{Z}}\boldsymbol{\delta}_k\right\|_2^2}$$

$$\leq 2\sqrt{2}\gamma\bar{f}_{\max}\alpha\sqrt{\ell(\boldsymbol{\xi}) - \mathcal{L}^\star}, \tag{84}$$

where $\bar{f}_{\max}$ is defined in (12) and the third line follows from Cauchy-Schwarz inequality.

