# OpenReview forum: "In-Context Learning with Representations: Contextual Generalization of Trained Transformers"
_NeurIPS.cc/2024/Conference — NeurIPS 2024 poster_

### Official Review · Reviewer_vG3D · 2024-07-08

**Soundness:** 2
**Presentation:** 3
**Contribution:** 3
**Rating:** 5
**Confidence:** 4

**Summary:**

This paper studies the training dynamics of multi-head Transformers by gradient descent on non-linear regression tasks via ICL. This work shows the linear convergence to the global minimum of the training and the in-context inference. An impressive contribution is that transformers are proven to learn contextual information to generalize to unseen examples when prompts contain a small number of input-label pairs. I am willing to update my review if my concerns are addressed.

**Strengths:**

1. The problem to be solved is significant, interesting, and challenging.
2. The paper is well-written.
3. The comparison with the related works makes the motivation clear.

**Weaknesses:**

1. It seems that the assumption that $H\geq N$ in Proposition 1 is too strong. The single-head works in Table 1 do not need this condition for few-shot ICL (Obviously, for single-head, $1<N$).  I know the problem to solve is different, but this means that Transformers cannot handle long contexts without a large number of heads.

2. No experiments are conducted to verify the theory. It is important to verify the theory by experiments, even with the most simplified Transformer model. For example, can you show if $H<N$, the learning will fail, while if $H\geq N$, the learning becomes successful by a one-layer Transformer?

3. Some statements related to the existing works are wrong. For example, in [Li et al., 2024], $N=\mathcal{O}(\epsilon^{-2}T)$ is the number of data for the pretraining to enable ICL for the Transformer. It is not the number of examples in each prompt as stated in line 129 of the submitted manuscript. The number of examples in each prompt in [Li et al., 2024] is $l\_{tr}$ and $l\_{ts}$ in Eqn 9 of Theorem 3.3.

4. I am unsure whether the theoretical proof is correct. For example, I don't know how you derive linear convergence of the loss via only PL condition and the smoothness of the loss function. Please see Question 1.

**Questions:**

1. Regarding Weakness 1, why do you have $\eta_Q$ in Eqn 63? I think by (41), (60), and Lemma 4.3 in [Nguyen and Mondelli, 2020], $\eta_Q$ should be replaced by $1$ in Eqn 63.

2. What is the requirement for the initialization of $w\_{h,k}$ defined in (7)?

3. It seems there is no requirement or assumption for the $f$ function to be learned. Is your bound derived for the worst case of all the possible $f$?

**Limitations:**

There is no potential negative societal impact of this work.

---

> ### Author Rebuttal · Authors · 2024-08-06
>
> ## Response to Reviewer vG3D
>
> Thank you for your insightful review. We've **added experiments to validate our theory in the supplementary pdf and our global response**. Below we address your concerns. If our responses resolve your questions, we'd highly appreciate your consideration in raising the score. Certainly, please don't hesitate to request any further clarification.
>
> > **(W1) Assumption $H\geq N$ too strong. The single-head works in Table 1 do not need this condition for few-shot ICL.**
>
> We recognize the importance of weakening this assumption and will explore this in future work. Meanwhile, we want to make the following clarifications.
>
> -  As explained in Section 3.3, it is generally impossible to train a single-head shallow transformer to succeed in the task we consider. Our task involves more complex non-linear regression with representation learning, which fundamentally requires multi-head attention.
>
> - Note that $H\geq N$ is a necessary and sufficient condition for the initialization condition (Assumption 3) to hold. To see why it is necessary, note that $B_k^{(0)}$ is an $N$-by-$H$ matrix, and has full row rank only when $H\geq N$.
>
> - Assumption 3 is crucial to our proof: it guarantees that $\zeta_0$ (defined in (12)) is positive, which in turn ensures $\sigma$ (the PL parameter in (44)) is positive.
>
> - However, it's important to note that Assumption 3 may not be a necessary condition for the convergence of the transformer. To weaken (or avoid) Assumption 3, other proof frameworks are needed. But due to the highly non-convexity of (even shallow) transformers, it's somewhat unavoidable to make other strong assumptions (see the examples we list in the next bullet point).
>
> - Though the single-head works in Table 1 do not need $H\geq N$, they often rely on other stronger assumptions even for tasks without representation learning, see Section 1 and 3.1 for details. To summarize:
>
>     - Huang et al. [2023], Li et al. [2024], Nichani et al. [2024] assume that the tokens are pairwise orthogonal, which implies the the token dimension $d$ is no smaller than the dictionary size $K$, while in practice $K\gg d$.
>
>     - Huang et al. [2023] and Chen et al. [2024] require the context length to be sufficiently large, so they are not analyzing *few-shot ICL*.
>
>     - Li et al. [2024] requires the width of their model to be larger than the square of the number of data.
>
>
>
> > **(W1) This means that Transformers cannot handle long contexts without a large number of heads/(W2) No experiments are conducted to verify the theory...can you show if $H<N$, the learning will fail, while if $H\geq N$, the learning becomes successful by a one-layer Transformer?**
>
> - Our above response indicates $H\geq N$ is only a sufficient (but may not be necessary) condition used to guarantee the convergence of the transformer. Therefore, our results don't indicate transformers cannot handle long contexts without a large number of heads in practice. However, shallow transformers have limited expressive power, which necessitates more attention heads to solve complex tasks. To support this, we have put empirical results in the supplementary pdf, please refer to Figure 3 and Figure 5 and the corresponding description in our global response.
>
>
>
>
> > **(W3) Statements on [Li et al., 2024] are wrong.**
>
> - You are correct, and we apologize for the mistake and will fix it in the updated paper. Thank you for your careful review.
>
>
> > **(W4/Q1)typo in (63)**
>
> - Thank you for your sharp observation. You are right -- $\eta_Q$ should be replaced by 1 in (63). We apologize for the typo and will fix it in the updated paper.
>
> - Let us emphasize that this typo does not affect the subsequent results, and the rest of our proof remain valid:
>
>     The revised (63) reads:
> $$
> \ell(\xi^{(t)})-\mathcal{L}^\star \leq \ell(\xi^{(t-1)})-\mathcal{L}^\star+\langle \nabla_\xi \ell (\xi^{(t-1)}), \xi^{(t)}-\xi^{(t-1)}\rangle +\frac{L}{2}||\xi^{(t)}-\xi^{(t-1)}||_2^2.
> $$
>     Combining this with (38), we have
> $$
> \ell(\xi^{(t)})-\mathcal{L}^\star\leq \ell(\xi^{(t-1)})-\mathcal{L}^\star + \eta_Q(L\eta_Q/2-1)||\nabla \ell (\xi^{(t-1)})||_F^2.
> $$
>
> When $\eta_Q\leq 1/L$, we have $L\eta_Q/2-1\leq -1/2$, which gives the first part of (64):
> $$\ell(\xi^{(t)})-\mathcal{L}^\star\leq \ell(\xi^{(t-1)})-\mathcal{L}^\star - \frac{\eta_Q}{2}||\nabla_\xi \ell (\xi^{(t-1)})||_F^2.$$
> Note that this is the only place where (63) is used.
>
> >**(W4) derive linear convergence of the loss via only PL condition and the smoothness of the loss function**
>
> - The combination of PL condition and smoothness is indeed sufficient for linear convergence of gradient descent. See for example Theorem 13.2 in this note: https://www.stat.cmu.edu/~siva/teaching/725/lec13.pdf
>
>
> - This approach is commonly used in deep learning optimization literature, as PL condition has been proven to hold for over-parameterized neural networks. See [1-3] for example.
>
> > **(Q2) initialization of $w_{hk}$**
>
> - $w_{hk}$ is initialized to be 0, as stated in line 226 of our paper.
>
> > **Is your bound derived for the worst case of all the possible $f$?**
>
> - our bounds apply for any possible $f$, which includes the worst case one. The key insight is that our theoretical guarantees depend on the function's outputs on the dictionary $\mathcal{V}$, i.e., in our analysis, we work with the finite-dimensional matrix $\widehat Z=(f_i(v_j))_{i\in[m],j\in[K]}\in R^{m\times K}$ defined in line 485, rather than the full functional form of $f$.
>
> ---
>
> [1] Liu, C., Zhu, L., and Belkin, M. (2020). Loss landscapes and optimization in over-parameterized non-linear systems and neural networks.
>
> [2] Karimi et al. (2016). Linear Convergence of Gradient and Proximal-Gradient Methods Under the Polyak-Łojasiewicz Condition.
>
> [3] Loizou et al. (2021). Stochastic polyak step-size for sgd: An adaptive learning rate for fast convergence.

---

> > ### Comment · Reviewer_vG3D · 2024-08-09
> >
> > Thank you for the reply. I am satisfied with the other answers except the one to W1. I still think the requirement that $H\geq N$ is too strong and not impractical. It is also not verified how loose the bound $H\geq N$ is by experiments. Can you at least verify by experiments that with a larger context length $N$, the number of heads $H$ should be larger to ensure a small loss?

---

> ### Author Response · Authors · 2024-08-09
> **Re: Reviewer vG3D**
>
> Thank you very much for your quick response! Below we change $N$ and find the smallest $H$ that achieves a training loss $\leq$ 0.5 within 100 iterations. We set $K=200, d=100, m=20, \tau=0.01$. Our result is as follows:
>
> | N   | 1   | 5   | 10  | 20  | 30  | 40  |
> |-----|-----|-----|-----|-----|-----|-----|
> | H   | 1   | 5   | 8   | 29  | 32  | 41  |
>
> This result verifies our theoretical finding that with a larger $N$, $H$ should be larger to ensure a small loss.
>
> If our response resolves your questions, we will highly appreciate your consideration in raising the score. Certainly, we are more than happy to answer your further questions.

---

> > ### Comment · Reviewer_vG3D · 2024-08-09
> >
> > Thank you for the response. I am wondering why you set the threshold of the training loss as 0.5. If you use a different threshold, will the bound still be tight? By the way, it seems that the result is not consistent with Figure 3 in the submitted PDF, where the case of $H=1$ cannot reach a loss of less than 0.5. Maybe the setting is different?

---

> > > ### Author Response · Authors · 2024-08-09
> > > **Re: Re: Reviewer vG3D**
> > >
> > > Thank you for your follow-up questions.
> > >
> > > >**regarding inconsistency with Figure 3**
> > >
> > > The configuration of the two settings are different. While both cases have $H=1$, in Figure 3, $N=30$ and clearly here $H=1$ cannot reach loss 0.5 (because our above experiment shows that $H$ should be at least 32 to reach loss 0.5 for $N=30$). But in the above experiment, $H=1$ is for $N=1$, where the loss can reach 0.5. We also mention that the parameters $K$ and $\tau$ are also set slightly different, but they are not the main reason to cause the difference.
> > >
> > > >**If you use a different threshold, will the bound still be tight?**
> > >
> > > We tried several configurations and found that the nature of the results are insensitive to the thresholds (and other hyperparameters): $N$ always increases along with $H$. In the following we present the results with loss threshold = 0.1 and 0.5 and iteration number = 200, and the other configurations remain the same as the above experiment:
> > >
> > >
> > > | N   | 1   | 5   | 10  | 20  | 30  | 40  |
> > > |-----|-----|-----|-----|-----|-----|-----|
> > > | H (threshold=0.5)  | 1   | 5   | 8   | 24  | 29  | 35  |
> > > | H (threshold=0.1)  | 1   | 7   | 21   | 36  | 51  | 57  |

---

> > > > ### Comment · Reviewer_vG3D · 2024-08-09
> > > >
> > > > Thank you for your response. Although I am still concerned that $H\geq N$ is not tight, the overall trend makes sense. I also like the convergence analysis of this work. I have increased my rating to 5.

---

> > > > > ### Author Response · Authors · 2024-08-13
> > > > >
> > > > > Many thanks for your feedback and increasing the score to support our paper. Regarding $H \ge N$, our numerical result somewhat indicates that $H$ cannot be much smaller than $N$ to guarantee a small loss. The main reason is because the one-layer transformer may not have substantial expressive power and hence could by nature requires a sufficiently large $H$. We suspect that such an assumption may not largely come from our technical tools, but may likely be due to the problem itself. It is great to hear you like the convergence analysis of this work, which we highly appreciate.

---

### Official Review · Reviewer_MzSs · 2024-07-11

**Soundness:** 3
**Presentation:** 3
**Contribution:** 3
**Rating:** 6
**Confidence:** 4

**Summary:**

This paper tackles the problem of task learning with representations using transformers. It presents a theoretical proof for convergence of training of a single-layer multi-head softmax attention transformer to the global minimum of the loss. Specifically, given $N$ example tokens and their corresponding task outputs, the transformer learns to predict the task output for unseen tokens $K-N$, where $K$ is the size of the dictionary. The paper shows that when trained for sufficiently long, the transformers can perform of ridge regression on the example tokens to label the unseen tokens.

**Strengths:**

(1) The paper shows that the parameters of a single-layer multi-head softmax attention transformers can be trained at linear convergence rate (Theorem 1) to the global minimizer of the training loss. This result elucidates that the remarkably efficient training that a transformer architecture offers.

(2) The paper proves that once trained for sufficiently long, the transformer performs ridge regression (Theorem 2) on the example tokens to label the query tokens during inference. This is a very promising step towards understanding the generalization capabilities of transformers, which is quite different from earlier works that require the number of example tokens $N$ to be large.

(3) The paper is written crisp and clean, making it very easy to follow.

**Weaknesses:**

(1) The optimal transformer configuration $(\boldsymbol{\theta})$ is not explicitly stated; only the functional form of the output is given, which makes it difficult to the interpret the result in the parameter $(\boldsymbol{\theta})$ space.

(2) The result in this paper is about the wide single-layer transformers (more heads), rather than the deep ones (more layers), and relies on Assumption 3, which holds when the number of heads, $H\ge N$ (Proposition 2). This is restrictive and there might be other ways in which the assumption can hold even if $N > H$. Some empirical demonstrations with shallow transformers on simple experiments might support this better.

**Questions:**

(1) Proposition 1 provides a way to make (almost) sure that Assumption 3 holds, but requiring $H \ge N$ seems very demanding in terms of compute. Can you comment on how to obtain similar results by simpler assumption?

(2) I am curios about the motivation behind your problem formulation, i.e., task-based labelling of a finite dictionary? Is there a real-world problem where this might be of use?

**Limitations:**

(1) One minor limitation is that the performance of the trained transformer depends on the relation between the number $N$ of example tokens and the dimension $m$ of the representational space (lines 295-299).

---

> ### Author Rebuttal · Authors · 2024-08-06
>
> ## Response to Reviewer MzSs
>
> Thank you for your time in reviewing our paper. Below We address your points. If our responses adequately address your concerns, we would be grateful if you could consider increasing your current score. And we are happy to answer your additional questions.
>
> > **(W1) The optimal transformer configuration $(\theta)$ is not explicitly stated; only the functional form of the output is given, which makes it difficult to the interpret the result in the parameter space.**
>
> - As is stated in line 316 and line 482-485, $\theta=${$Q_h,w_h$} is optimal if and only if it satisfies:
> $$
> \forall k: \sum_{h=1}^H w_{h,k} softmax(V^\top Q_h v_k)=A_{:,k},$$
> where matrix $A$ is defined in (33).
>
>     Note that due to the non-linear nature of the softmax function and the complex interplay between Q_h and w_h, an explicit closed-form solution in the parameter space is not feasible.
>
> - This characterization, while not explicitly in the parameter space, shows that the transformer gains in-context ability by learning to approximate the matrix $A$, which encodes the "inherent information" of the tasks (basic function maps), through a combination of its attention mechanism (represented by the softmax terms) and output weights. It's worth noting that this characterization shows the role of multi-head attention in approximating complex functions (see line 330-339) and connects the learned parameters to the underlying structure of the tasks.
>
> > **(W2/Q2) Assumption 3 and the $H \geq N$ condition**
>
> We recognize the importance of potentially weakening this assumption and will explore this in future work. Thank you for raising this. Meanwhile, we also want to make the following clarifications.
>
> - To weaken (or avoid) Assumption 3, other proof frameworks are needed. For example, one possible way is to extend the analysis in (Huang et al. 2023) [9] based on induction hypothesis. However, due to the highly non-convexity of (even shallow) transformers, it's somewhat unavoidable to make other strong assumptions. For examples, similar works such as those listed in Table 1 make stronger assumptions than us despite they consider simpler tasks, see Section 1 for details.
>
> - While the above argument indicates $H\geq N$ may not be necessary to guarantee the convergence of the transformer, it's intuitive that $H$ may need to be large in our setting. This is because shallow transformers have limited expressive power, necessitating more attention heads to solve complex tasks. In fact, we have shown that it's generally impossible to train a transformer to succeed in the non-linear regression tasks we consider at least when $H=1$, see line 330-339 for details.
>
> - The experiment in Figure 3 in the supplementary pdf shows that when $H<N(=30)$, the loss stopped descending when it's far from the minimal value. And it keeps descending when $H=N$ (though slowly).
>
> > **(Q2) motivation/real-world applications**
>
> Thank you for your question. We'll incorporate the following in our updated paper:
>
> - The non-linear regression setting is quite general and can model a wide range of real-world problems across various domains, including classification (by using a softmax function as the output layer [1]) time series forecasting [2], inverse problems[3], financial modeling [4], and physical/mathematical modeling [5].
>
> - A key aspect of our work is the analysis of how transformers learn representations (see Section 3.3). This is fundamental to many complex machine learning tasks, including feature extraction, translation and generalization. Our insights into how transformers extract and memorize "inherent information" of basic function maps during training (lines 312-317) could be valuable for understanding these more complex tasks. Besides, our novel insights into how transformers acquire contextual generalization ability with limited data and handle underdetermined templates is relevant to a broader range of ICL tasks beyond regression.
>
> - While our analysis uses a finite dictionary, this approach allows us to derive rigorous theoretical results while still capturing key aspects of how transformers learn and generalize. The insights gained from this analysis can inform our understanding of transformer behavior in more general settings.
>
> > **(limitation) the performance of the trained transformer depends on the relation between $m$ and $N$**
>
> - You're correct that the inference-time performance depends on the $m$-$N$ relationship. This behavior highlights a fundamental trade-off in ICL between having enough examples to determine the underlying function and avoiding overfitting to noisy labels. Far from being a limitation, this is a key finding of our work and reveals nuanced behavior in ICL that differs from traditional supervised learning.
>
> - We've conducted experiments to verify this finding, please refer to Figure 2 in the supplemental pdf and the corresponding descriptions in our global response.
>
>
>
>
> ---
>
> [1] Liu, C., Zhu, L., and Belkin, M. (2020). Loss landscapes and optimization in over-parameterized non-linear systems and neural networks.
>
> [2] Karimi et al. (2016). Linear Convergence of Gradient and Proximal-Gradient Methods Under the Polyak-Łojasiewicz Condition.
>
> [3] Loizou et al. (2021). Stochastic polyak step-size for sgd: An adaptive learning rate for fast convergence.
>
> [4] Fischer, Manfred M. (2015). Neural networks. A class of flexible non-linear models for regression and classification.
>
> [5] F Stulajter (2002). Predictions in Time Series Using Regression Models.
>
> [6] R Nickl (2021). On some information-theoretic aspects of non-linear statistical inverse problems.
>
> [7] T Amemiya (1983). Non-linear regression models.
>
> [8] M Alizamir et al. (2020). A comparative study of several machine learning based non-linear regression methods in estimating solar radiation: Case studies of the USA and Turkey regions.
>
> [9] Y Huang et al. (2023). In-Context Convergence of Transformers.

---

> > ### Comment · Reviewer_MzSs · 2024-08-12
> >
> > Thank you for the valuable comments, and for the clarification on the limitation (the relation between $m$ and $N$ affecting the performance) to be a strength of the paper. I agree with you.
> >
> > Despite the clean proof and the result of the fast convergence of the training, the finite dictionary still seems to be a limitation as the transformer can essentially 'memorize' the basis vectors into the value transforms in the heads. As such, the paper is more about how an attention mechanism can 'memorize' efficiently (possibly better than other architectures), which I agree to be novel. However, direct impact of the paper is unclear. Moreover, the construction using a large number of heads is not very appealing.
> >
> > Taking all of these into consideration, I shall keep my (positive) score.

---

> ### Author Response · Authors · 2024-08-12
>
> Dear Reviewer MzSs,
>
> We've taken your initial feedback into careful consideration in our response. Could you kindly confirm whether our responses have appropriately addressed your concerns?
>
> If you find that we have properly addressed your concerns, could you please kindly consider increasing your initial score accordingly? Please let us know if you have further comments.
>
> Thank you for your time and effort in reviewing our work!
>
> Many thanks, Authors

---

> ### Author Response · Authors · 2024-08-12
>
> Dear Reviewer MzSs,
>
> Thank you very much for your response!
>
> In addition to what you mentioned as 'memorize' the basis vectors, our results contain two more points: (i) after 'memorizing/learning' basis vectors, the ICL implements **ridge regression** to use such basis information, which is a new characterization; (ii) such tranformers having those properties (memorizing basis vectors and implementing ridge regression) can be learned naturally by gradient descent.
>
> Many thanks,
> Authors

---

### Official Review · Reviewer_TyrV · 2024-07-12

**Soundness:** 2
**Presentation:** 2
**Contribution:** 3
**Rating:** 4
**Confidence:** 2

**Summary:**

The paper investigates the theoretical understanding of in-context learning, focusing on whether transformers can generalize to unseen examples within a prompt by acquiring contextual knowledge. The paper analyzes the training dynamics of transformers using non-linear regression tasks, demonstrating that multi-head transformers can predict unlabeled inputs given partially labeled prompts. The study shows that the training loss for a shallow multi-head transformer converges linearly to a global minimum, effectively learning to perform ridge regression. Main contributions include: (1) The paper establishes the convergence guarantee of a shallow transformer with multi-head softmax attention trained with gradient descent on general non-linear regression in-context learning tasks. (2) The paper analyzes the transformer’s behavior at inference time after training, demonstrating that the transformer decides its generating template by performing ridge regression. (3) The analysis framework allows overcoming several assumptions made in previous works, such as the requirement for large prompt lengths, orthogonality of data, restrictive initialization conditions, special structure of the transformer, and super wide models.

**Strengths:**

1. The focus on analyzing training dynamics and convergence guarantees for transformers in ICL tasks is relatively novel. Prior works often concentrate on empirical performance without delving deeply into the theoretical underpinnings.
The combination of gradient descent analysis with multi-head softmax attention in the context of non-linear regression is a creative fusion of ideas that opens up new avenues for understanding how transformers learn contextually.
By overcoming assumptions such as the need for large prompt lengths, orthogonality of data, and super wide models, the paper advances the state of the art and provides a more realistic framework for understanding transformer performance.

2. The paper’s analysis of the convergence dynamics and ridge regression behavior of transformers is mathematically robust, with detailed proofs and clear logical progression.

3. The paper is well-organized, with a logical flow from the problem statement to the theoretical analysis and conclusions, making it easy to follow the argumentation.

**Weaknesses:**

While the paper excels in theoretical analysis, it lacks empirical validation of the proposed concepts and results. It is suggested to include empirical experiments that demonstrate the practical applicability of the theoretical findings would significantly strengthen the paper. For instance, running experiments on standard benchmarks to show how the proposed theoretical insights translate to improved performance in real-world tasks would validate the results more strongly.

The paper’s discussion on the practical implications of the theoretical findings is somewhat limited. This can make it challenging for practitioners to understand how to apply these insights to real-world problems.

**Questions:**

1. Do you have any empirical results or planned experiments that demonstrate the practical applicability of your theoretical findings?

2. What are the potential impacts of the remaining assumptions and simplifications (e.g., specific initialization conditions, structure of the transformer) on the generalizability of your results?

**Limitations:**

Please review the weaknesses.

---

> ### Author Rebuttal · Authors · 2024-08-06
>
> ## Response to Reviewer TyrV
>
> Thank you for your valuable feedback. We agree that empirical validation is crucial and **have included experimental results in the supplementary PDF and the global response**. Note that doing in-context training on real datasets is highly demanding and usually requires very large models trained on a lot of different datasets [3]. Therefore, it's standard practice for the line of works (e.g., [4,5]) that analyze transformers' in-context ability to experiment on the synthetic dataset.  Below we address your other points. If these clarifications resolve your primary concerns, we would highly appreciate your consideration of increasing your score. Certainly, please don't hesitate to request any further clarification.
>
>
> >**(Weaknesses/Q1) discussion on the practical implications of the theoretical findings**
>
> Besides those discussed in the global response, one interesting implication is that our paper suggests assigning different learning rates for different parameter blocks, which has been shown to work well by some very recent work (e.g. [1,2]), where the authors show sgd (which assigns the same learning rate to all blocks) doesn't work well for transformers, but it's also not necessary to keep different learning rates for all parameters as Adam does.
>
>
>
> >**What are the potential impacts of the remaining assumptions and simplifications (e.g., specific initialization conditions, structure of the transformer) on the generalizability of your results?**
>
> Thank you for bringing up this important question. We'll incorporate the following discussion in our paper:
>
> - While our current analysis focuses on shallow transformers, we believe our methodology has the potential to be extended to deeper architectures. The core of our proof technique, which combines the Polyak-Łojasiewicz (PL) condition with smoothness to demonstrate convergence (as detailed in the proof of Theorem 1), could potentially be generalized to deeper transformers. Therefore, we expect that our key results, such as the limit point of the output of the transformer (see (19) in Theorem 2), the explanation of the transformer's ICL ability and the analysis of its generalization ability would also apply to deep Transformers.
>
> -  The main challenge that we need to overcome in extending the proofs to deeper architectures lies in estimating the PL coefficient $\sigma$ and smoothness coefficient $L$. To this end, it is possible to leverage our techniques for one-layer transformer and carefully conduct a block-by-block analysis potentially via induction arguments.
>
> - As we comment in line 217-218 and Proposition 1 in the paper, our initialization condition can be easily achieved via random sampling from Gaussian distribution.
>
> - It's worth noting that we assume Gaussian noise for simplicity and show that the transformer decides its generating template by performing ridge regression (see Section 1.1 and Theorem 2). Our technique could potentially be extended to analyze other types of noise. For example, we could hypothesize that if the noise is Laplacian, the transformer might learn to perform Lasso regression instead. We leave this exploration as future work.
>
> ---
>
> [1] Y Zhang et al. (2024). Adam-mini: Use Fewer Learning Rates To Gain More.
>
> [2] Y Zhang et al. (2024). Why Transformers Need Adam: A Hessian Perspective.
>
> [3] S Min (2022). MetaICL: Learning to Learn In Context.
>
> [4] S Garg et al. (2022). What Can Transformers Learn In-Context? A Case Study of Simple Function Classes.
>
> [5] S Chen et al. (2024). Training Dynamics of Multi-Head Softmax Attention for In-Context Learning: Emergence, Convergence, and Optimality.

---

> ### Author Response · Authors · 2024-08-10
>
> Dear Reviewer TyrV,
>
> We've taken your initial feedback into careful consideration in our response. Could you kindly confirm whether our responses have appropriately addressed your concerns?
>
> If you find that we have properly addressed your concerns, could you please kindly consider increasing your initial score accordingly? Please let us know if you have further comments.
>
> Thank you for your time and effort in reviewing our work!
>
> Many thanks, Authors

---

> > ### Author Response · Authors · 2024-08-12
> > **kind reminder as the rebuttal deadline approaches**
> >
> > Dear Reviewer TyrV,
> >
> > As the author-reviewer discussion period will end soon, we would like to check whether our responses have properly addressed your concerns? If so, could you please kindly consider increasing your initial score accordingly? Certainly, we are more than happy to answer your further questions.
> >
> > Thank you for your time and effort in reviewing our work!
> >
> > Best Regards,
> > Authors

---

> > > ### Author Response · Authors · 2024-08-13
> > >
> > > Dear Reviewer TyrV,
> > >
> > > The author-reviewer discussion period will end in less than one day. We sincerely hope to receive your feedback and see if our responses have properly addressed your concerns. If so, could you please kindly consider increasing your initial score accordingly? Certainly, we are more than happy to answer your further questions.
> > >
> > > Thank you for your time and effort in reviewing our work!
> > >
> > > Best Regards, Authors

---

### Official Review · Reviewer_dubr · 2024-07-29

**Soundness:** 3
**Presentation:** 3
**Contribution:** 2
**Rating:** 7
**Confidence:** 4

**Summary:**

This paper presents a rigorous theoretical analysis of the training dynamics and generalization capabilities of a one-layer transformer with multi-head softmax attention for in-context learning (ICL) of non-linear regression tasks. The authors consider a more challenging and realistic setting where prompts contain only a small number of noisy labeled examples, insufficient to fully determine the underlying template function. This addresses limitations in previous theoretical work that often required unrealistically large numbers of examples per prompt.

A key theoretical contribution is proving convergence of the training loss to its global minimum at a linear rate when optimizing with gradient descent. This is the first such convergence result for transformers with multi-head softmax attention on ICL tasks. The analysis reveals that the transformer effectively learns to perform ridge regression during training. The paper provides precise bounds on the mean squared error between the transformer's predictions and those of the optimal ridge regression solution, as a function of the number of training iterations.

The authors demonstrate that multi-head attention is crucial for succeeding at this ICL task. They provide a lower bound on the number of heads needed (H ≥ N), while also noting that too many heads can slow down convergence. The analysis shows how the multi-head mechanism allows the transformer to approximate the required matrix operations for ridge regression.

**Strengths:**

- Setting : The paper overcomes several limiting assumptions made in previous theoretical work, such as requiring very long prompts, orthogonal input data, restrictive initialization conditions, or unrealistically wide models. Key technical innovations include a novel reformulation of the loss function to remove expectations, proving smoothness and the Polyak-Łojasiewicz condition for the loss landscape, and carefully bounding various matrix norms and eigenvalues throughout training.

- Intuition and connection to ridge regression : The analysis provides insight into how the regularization strength in the implicit ridge regression depends on the ratio of the feature dimension m to the number of examples N in each prompt. Also, the authors show that the transformer acquires two types of generalization capabilities: contextual generalization to unseen examples within a task by learning to infer and apply the underlying template function, and generalization to unseen tasks by learning a general strategy (ridge regression) that works across different λ vectors.

The paper provides rigorous mathematical analysis in a realistic setting. In doing so it offers novel insights, in particular the notion of implicitly learning to perform ridge regression and acquire generalizable knowledge about the representation function.

Also, the paper provides the first convergence guarantees for transformers with multi-head softmax attention on ICL tasks, which is a significant theoretical contribution.

**Weaknesses:**

- Limited model architecture (one-layer transformer) which is understandable

- Lack of empirical validation:  The paper is purely theoretical and does not include any experimental results to validate its predictions.

- Focus on regression : The analysis is limited to regression tasks, which represent only a subset of the problems typically addressed by transformers and in-context learning. Many real-world applications involve classification, sequence generation, or more complex structured prediction tasks. It's not immediately clear how the insights about ridge regression would translate to these other task types.

Lack of coomparison to other ICL approaches: The paper doesn't provide a comprehensive comparison to other theoretical approaches to in-context learning. While it does mention some previous work, a more in-depth discussion of how this approach relates to or improves upon other theoretical frameworks for ICL would have provided valuable context.

**Questions:**

Have the authors considered adding any toy domain experiments that would illustrate and further validate their claims ?

**Limitations:**

- Simplified model: The analysis is limited to a one-layer transformer, which is much simpler than state-of-the-art models used in practice.
- Lack of empirical validation: While the theoretical results are impressive, the paper doesn't include experimental results to validate its predictions.
- Focus on regression: The paper only considers regression tasks, and it's not immediately clear how these results would generalize to classification or more complex tasks.

---

> ### Author Rebuttal · Authors · 2024-08-06
>
> ## Response to Reviewer dubr
>
> Thank you for your review and positive comments. We've **included some experiments to verify our theoretical findings in the supplementary pdf and the global response**. Below we address your other points. If our responses resolve your concerns, we'd highly appreciate your consideration of increasing your current score. Certainly, please also let us know if you have further questions.
>
> >**simplified model**
>
> - Our analysis indeed focuses on shallow transformers, but we believe our methodology lays a foundation for extending to deeper architectures. The core of our proof technique, which combines the Polyak-Łojasiewicz (PL) condition with smoothness to demonstrate convergence (as detailed in the proof of Theorem 1), provides a framework that could potentially be adapted for deeper transformers. We anticipate that our key results, such as the limit point of the transformer's output (see (19) in Theorem 2), the explanation of the transformer's ICL ability, and the analysis of its generalization capability, would extend to deeper models, albeit with more complex derivations.
>
> - However, extending the proofs to deeper architectures presents significant challenges, particularly in estimating the PL coefficient $\sigma$ and smoothness coefficient $L$. As evident from our proofs, even for a one-layer transformer, this computation is highly non-trivial and requires a meticulous block-by-block analysis. For deeper models, this analysis would become substantially more intricate, requiring novel mathematical techniques to handle the increased complexity.
>
> - To the best of our knowledge, there's no existing work analyzing deep transformers' training dynamics from an optimization perspective. Most theoretical works, including those in Table 1, focus on shallow transformers due to these challenges. Developing rigorous analytical methods for deep transformers remains an open and exciting direction for future research in this field.
>
> >**focus on regression**
>
> - Our framework considers non-linear regression tasks, which are significantly more complex than the linear tasks analyzed in previous theoretical works (as highlighted in Table 1 of our paper). We expect that such a contribution within regression framework is valuable to the community.
>
> -   A key aspect of our work is the analysis of how transformers learn representations (see Section 3.3). This is fundamental to many other machine learning tasks, including feature extraction, translation, and generalization. Our insights into how transformers extract and memorize "inherent information" of basic function maps during training (lines 312-317) could be valuable for understanding other more complex tasks. Besides, our novel insights into how transformers acquire contextual generalization ability with limited data and handle underdetermined templates is relevant to a broader range of in-context learning tasks beyond regression. We leave the extensions of our settings as important directions for future work.

---

> ### Author Response · Authors · 2024-08-12
>
> Dear Reviewer dubr,
>
> We've taken your initial feedback into careful consideration in our response. Could you kindly confirm whether our responses have appropriately addressed your concerns?
>
> If you find that we have properly addressed your concerns, could you please kindly consider increasing your initial score accordingly? Please let us know if you have further comments.
>
> Thank you for your time and effort in reviewing our work!
>
> Many thanks, Authors

---

> > ### Comment · Reviewer_dubr · 2024-08-14
> >
> > Thanks for your rebuttal and the additional toy experiments, which I find sensible (as exemplified by discussion with Reviewer vG3D) and illustrative. I have therefore increased my score.

---

### Author Rebuttal · Authors · 2024-08-06

We conducted the following experiments to validate our theoretical findings. The attached file includes the experimental plots. We will add these results as an experiment section in our revision.

**Set up.** We conduct experiments using synthetic dataset (which is standard practice of this line of works that analyze transformers' in-context ability, see [1,2] for example), where we randomly generate each token $v_k$ and their representation $f(v_k)$ from standard Gaussian.  We experiment either on the 1-layer transformer described in Section 2 or a standard 4-layer transformer in [3] with $d_{model}=256$ and $d_{ff}=512$. For the 1-layer transformer experiments, we set the training loss to be the population loss defined in (8), and initialize $Q_h^{(0)}$ using standard Gaussian and set $w_h^{(0)}$ to be 0 ($h\in[H]$), identical to what is specifed in Section 3. For the experiments on 4-layer transformers, we generate $\lambda$ from standard Gaussian distribution to create the training set with 10000 samples and in-domain test set with 200 samples; we also create an out-of-domain test set with 200 samples by sampling $\lambda$ from $N(1e,4I)$. Given $\lambda$, we generate the label $y_k$ of token $v_k$ using (1), $k\in[K]$. We train with a batch size 256. All experiments use the Adam optimizer with a learning rate 1e-4.

Figure 1 in the attached file shows the training and inference loss of the 1-layer transformer, where we measure the inference loss by $\frac{1}{K}||\hat y-\hat y^\star||_2^2$ to validate (19): after sufficient training, the output of the transformer $\hat y$ converges to $\hat y^\star$. We set $N=30$, $K=100$, $d=100$, $m=20$, $H=30$. We generate $\lambda$ from $N(0.5e,0.01I)$, which is out of the training distribution (Assumption 1). Thus this experiment also shows the transformer's contextual generalization to unseen examples and generalization to unseen tasks, validating our claim in Section 3.2. We also validate the two types of generalization capabilities on the 4-layer transformer, as shown in Figure 4, where we set $K=200$ and the other configurations same as in Figure 1. From Figure 4 we can see that the three curves have the same descending trend, despite the inference loss on the ood dataset is higher than that on the in-domain dataset.

Figure 2 verifies our claim at the end of Section 2: the closer $m$ is to $N$, the better the transformer's choice is. To be specific, we fix $m=100,\tau=0.01$, and plot $\frac{1}{K}||\hat{y}^\star-\hat{y}^{\text{best}}||_2^2$ corresponding to $N$ from 50 to 150. When $N=m=100$, this square norm becomes exactly 0.

Figure 3 validates our claim on the number of attention head $H$ at the end of Section 3.3 using the 1-layer transformer. In this experiment we use different $H$ to plot the training loss curves, and set the other configurations same as those in Figure 1.  From Figure 3 we can see that we need to set $H$ large enough to guarantee the convergence of the training loss. However, setting $H$ too large ($H=400$) leads to instability and divergence of the loss. Recall that in Proposition 1, we require $H\geq N$ to guarantee our convergence results hold. Although this condition may not be necessary, Figure 3 shows that when $H<N=30$, the loss stopped descending when it's far from the minimal value. On the other side, the loss keeps descending when $H=30$ (though slowly).

We also explore how $H$ affects the training on the 4-layer transformer, as displayed in Figure 5, where we set $K=200$ and the configurations other than $H$ are the same as in Figure 3. We fix the wall-clock time to be 100 seconds and plot the training loss curves with different $H$. The left figure shows the final training and losses change with $H$. It reflects that the losses converge faster with smaller $H$ (here the final training loss is the smallest when $H=4$). The right figure of training curves corresponding to different $H$ within 100s may provide some explanation to this phenomenon: (i) transformers with larger $H$ could complete less iterations within a fixed amount of time (the curves corresponding to larger $H$ are shorter); (ii) the training loss curves corresponding to large $H$ ($H=32,64$) descend more slowly. This indicates our claim that larger $H$ may yield slower convergence rate is still valid on deeper transformers. Note that unlike the 1-layer transformer, deeper transformers don't require a large $H$ to guarantee convergence. This is because deep transformers have great expressive power even when $H$ is small.

---

[1] S Garg et al. (2022). What Can Transformers Learn In-Context? A Case Study of Simple Function Classes.

[2] S Chen et al. (2024). Training Dynamics of Multi-Head Softmax Attention for In-Context Learning: Emergence, Convergence, and Optimality.

[3] A Vaswani et al. (2017). Attention is all you need.

---

### Decision · Program_Chairs · 2024-09-25

**Decision:**

Accept (poster)

**Comment:**

This paper provides a theoretical account of in-context learning in a single-layer transformer performing nonlinear regression tasks. This moves beyond prior theoretical accounts along some dimensions — though assumptions like a single layer are still quite limiting. The reviewers generally agreed that the paper offered a useful contribution despite these limitations, especially after clarification in the rebuttal period, but that the papers impact might be limited because of both its strong assumptions, and the relative lack of empirical validation or further practical insight provided.